# On the Complexity of Neural Computation in Superposition

## Abstract

Superposition, the ability of neural networks to represent more features than neurons, is increasingly seen as key to the efficiency of large models. We investigate the theoretical foundations of computing in superposition. We present the first lower bounds for a neural network computing in superposition, showing that for a broad class of problems, including permutations and pairwise logical operations, computing $m'$ features in superposition requires at least $\Omega(\sqrt{m' \log m'})$ neurons and $\Omega(m' \log m')$ parameters. This implies an explicit limit on how much one can sparsify or distill a model while preserving its expressibility, and complements empirical scaling laws by implying the first subexponential bound on capacity: a network with $n$ neurons can compute at most $O(n^2 / \log n)$ features. Conversely, we provide a nearly tight constructive upper bound: logical operations like pairwise AND can be computed using $O(\sqrt{m'} \log m')$ neurons and $O(m' \log^2 m')$ parameters. There is thus an exponential gap between the complexity of computing in superposition (the subject of this work) versus merely representing features, which can require as little as $O(\log m')$ neurons based on the Johnson-Lindenstrauss Lemma. Our work analytically establishes that the number of parameters is a good estimator of the number of features a neural network computes.

## 1 Introduction

While neural networks achieve remarkable empirical success across diverse domains, understanding the computational principles and representations underlying their decision-making processes remains a fundamental challenge. Recent groundbreaking work on this problem of *mechanistic interpretability* (Bricken et al., 2023; Elhage et al., 2022b; Olah et al., 2020; Templeton et al., 2024) has demonstrated that *features*, functions recognizing specific input properties, form fundamental computational units of neural networks. Features may represent concrete objects, abstract ideas, or intermediate computational results. For instance, (Templeton et al., 2024) identified approximately twelve million human interpretable features in the Claude 3 Sonnet model, including a notably robust "Golden Gate Bridge" feature activating across multiple contexts, languages, and modalities.

The main challenge with extracting these features is that networks typically utilize many more features than available neurons. When a network uses more features than neurons, it is said to be computing in *superposition* (Arora et al., 2018b; Elhage et al., 2022a;b; Olah et al., 2020) as opposed to *monosemantic* computation, which has a one-to-one feature to neuron mapping. This concept was popularized by (Elhage et al., 2022b), which introduced the *superposition hypothesis*: neural network training leads to a representation of features using nearly-orthogonal feature vectors in neuron activation space, which allows the network to represent more features than neurons. Neuronal activation space vectors are defined by the activation values of the neurons at a given layer of the network on the computation of a specific input. Thus, feature vectors can be seen as an encoding of which features are active. Note that if a neural network is computing in superposition, then it must also be using *polysemantic* representations (Arora et al., 2018b; Chan, 2024), where neurons participate in the representation of more than one feature.

Superposition is important for computational efficiency, as large models appear to employ at least hundreds of millions features and quite likely orders of magnitude more (Templeton et al., 2024), making monosemantic representations infeasible without a significant increase in the size of the model (Elhage et al., 2022b; Olah et al., 2020). However, superposition significantly complicates

interpretability and explainability, as the actual number of features utilized by models remains poorly understood. For example, in hypothesizing about how complete the set of twelve million of Claude 3 Sonnet's features was, (Templeton et al., 2024) stated "We think it's quite likely that we're orders of magnitude short." Furthermore, by the Johnson-Lindenstrauss Lemma (Johnson & Lindenstrauss, 1984), the "nearly orthogonal vector" representation may allow the number of features to be exponential in the number of neurons. Central to this and the superposition hypothesis more broadly is the assumption of *feature sparsity*, the empirical observation that only a small subset of features is active during any computation (Chan, 2024; Elhage et al., 2022b; Henighan et al., 2023; Olah et al., 2020; Templeton et al., 2024).

Much work on feature superposition in neural networks has concentrated on the *representation* problem: how are the features encoded within a trained model? Sparse Autoencoders and related techniques (Bricken et al., 2023; Cunningham et al., 2023; Dunefsky et al., 2024; Elhage et al., 2022b; Gao et al., 2024; Rajamanoharan et al., 2024; Templeton et al., 2024) have been shown to learn how specific features are encoded in specific instances of trained networks. In contrast to this focus on representation, our work investigates *computation* with superposed features. We address the fundamental question: Given the logic connecting a set of features, how can a neural network implement this logic in superposition? Specifically, we provide theoretical limits of superposition efficiency by identifying the minimum number of neurons and parameters needed for such computations.

Our work abstracts the challenge of "computation with superposed features" into a classic algorithmic framework. This approach provides a two-pronged contribution to the science of mechanistic interpretability. First, our lower bounds establish fundamental information-theoretic limits on how efficiently features can be computed in parallel, addressing the open question of how many features a given number of neurons can process. Second, our algorithmic constructions offer concrete, testable hypotheses about the mechanisms neural networks might use to operate efficiently within these boundaries. As described below, one of these mechanisms has been subsequently found in (Adler et al., 2025) to appear in traditionally trained small neural networks.

To make this problem tractable while retaining its core challenges, we follow (Vaintrob et al., 2024) and focus on the parallel computation of multiple Boolean functions, specifically multiple k-wise ANDs. This abstraction serves three key purposes. First, it distills the immense complexity of general neural computation into a well-defined, analyzable model that captures the essential challenge of managing the interference patterns that arise in superposition. Second, the parallelism in our problem directly mirrors a key, but poorly understood, aspect of neural computation, where numerous features can be processed simultaneously by the network. Finally, this formalization makes the problem amenable to algorithmic and complexity-theoretic tools, allowing us to derive rigorous bounds and explicit constructions. Through this focused approach, we aim to provide foundational insights into the computational principles governing general neural computation. It is also possible that Boolean logic forms a fundamental building block of more complex neural computations, for example in domains like computer vision that rely on detecting specific combinations of hierarchically organized features.

**Problem Formulation:** We consider a neural network tasked with computing a collection of Boolean formulas in parallel. Formally, let $F = (f_1, \ldots, f_{m'})$ be a set of $m'$ Boolean formulas, each defined over $m$ input variables. Let $U \subseteq \{0, 1\}^m$ be a set of admissible inputs (where each $u \in U$ is an instantiation of the $m$ Boolean variables). Our goal is to construct a neural network $N(F)$ such that, for every $u \in U$, the network computes $(f_1(u), \ldots, f_{m'}(u))$ (possibly with errors).

We view each input $u$ as identifying which features are active as the input to one logical layer of a neural network. The formulas in $F$ then specify a logical mapping that determines the new set of active features for the subsequent logical layer. In a trained network, this logical mapping might be computed by a single or small number of physical layers. Multiple such logical layers can then be chained together to implement the overall neural computation.

A class $\mathcal{F}$ of problems $F$ we study here (and then generalize), is 2-AND, introduced by (Vaintrob et al., 2024), consisting of all ordered sets of $m'$ pairwise ANDs of $m$ variables. For a fixed $m$ and $m'$, let $\mathcal{F}_{m,m'}$ be the class of all $F = (f_1, \ldots, f_{m'})$, where each $f_i$ is an AND of two out of the $m$ variables. Note that $|\mathcal{F}_{m,m'}| = \binom{\binom{m}{2}}{m'} m'!$ We refer to 2-AND restricted to a specific $m$ and $m'$ as 2-AND$_{m,m'}$. We also study the *Neural Permutation* problem, where $m' = m$ and $\mathcal{F}$ is the set of all permutations of the identity function on $m$ Boolean inputs.

Our primary measure of complexity is $n$, the number of neurons in $N(F)$. We also are interested in the total parameter count of $N(F)$ which is a function of $m$. A model is said to compute in superposition if $n < m$. We here consider the scenario where both the input and output of the layer are in superposition, and thus the $m$ inputs need to be represented with $n$ dimensions instead of $m$. This is fundamentally enabled by the principle of feature sparsity: for any given computation, only a small fraction of the total features are active (Elhage et al., 2022b; Templeton et al., 2024). This sparsity is a prerequisite for superposition; without it, the nearly-orthogonal vectors that encode features would suffer from excessive interference, making it impossible to reliably represent more features than neurons. Accordingly, feature sparsity is both an empirical reality observed in large models and a foundational assumption in all work on superposition (Chan, 2024; Henighan et al., 2023; Olah et al., 2020; Vaintrob et al., 2024).

Reflecting this principle, we impose a feature sparsity constraint on the set of admissible inputs $U$. We say $U$ is *feature sparse* $v$ if every $u \in U$ has at most $v$ ones. When $v \ll m$, this condition enables a compressed, superposed representation for both the $m$ input variables and the $m'$ outputs. We define our models of computation in more detail in Appendix B.

**Our Results** This paper establishes nearly tight bounds on the resources required for computing several functions in superposition: Neural Permutation, 2-AND, and generalizations of 2-AND. We show that using superposition allows $m'$ features to be computed using approximately $\sqrt{m'}$ neurons, but further compression is not possible.

**Lower Bounds:** Our lower bounds (Appendix D) hold for a general computational model (defined in Appendix B), that encompasses neural networks irrespective of specific architectural details like activation functions or connectivity patterns. We introduce a general technique within this model, and then use this technique to show that the minimum description of the parameters of the neural network must be at least $\Omega(m' \log m')$ bits, for a broad class of problems that includes Neural Permutation, as well as 2-AND$_{m,m'}$. For neural networks in our upper bound model using a constant number of bits per parameter, this implies $\Omega(\sqrt{m' \log m'})$ neurons are required.

This lower bound applies to both perfectly accurate networks and those that can have some errors. For networks that must always be correct, the proof is a simple counting argument. The analysis becomes slightly more involved for the case where errors are permitted—reflecting what happens in practice. Our proof leverages the network's *expressibility*: the diversity of functions computable by varying its parameters. Using Kolmogorov Complexity, we prove that high expressibility necessitates a large parameter description length, even allowing for errors. We also show that a similar, slightly weaker, lower bound holds even if the network is only required to compute the correct set of outputs, without regard to their order. In this 'unordered' case, the parameter description length must be at least $\Omega(m' \log \frac{m(m-1)}{2m'})$ bits for the case of no errors, and $\Omega(m')$ bits with errors allowed, provided that $m' \leq \binom{m}{2}/2$. All our lower bounds hold for inputs with feature sparsity 2 and are information-theoretic. They require no structural assumptions about the network, unlike typical VC dimension based lower bounds (Vapnik & Chervonenkis, 1971) on neural networks.

Our lower bounds have important implications to the study of mechanistic interpretability. No prior evidence suggested that the number of features must be less than exponential in the number of neurons when using superposition. Our lower bounds on neurons imply the first subexponential upper bound on the number of features that can be computed. Specifically, for the problems we consider, a network (or network layer) with $n$ neurons can only compute $O(n^2/\log n)$ output features. The lower bounds also contrast sharply with passive representation (encoding active features without computation), where techniques like the Johnson-Lindenstrauss Lemma (Johnson & Lindenstrauss, 1984) or Bloom filters (Bloom, 1970; Broder & Mitzenmacher, 2002) allow $n$ neurons to represent up to $2^{O(n)}$ features. Our results therefore show an exponential gap between the capacity for passive representation and active computation in superposition.

The lower bounds also have implications for model compression, a topic of great interest given the memory limitations of today's GPUs (Hoefler et al., 2021). Compression techniques used in practice include *quantization* (Hoefler et al., 2021), *sparsity* (Frankle & Carbin, 2018; Hoefler et al., 2021), and *knowledge distillation* (Hinton et al., 2015). Our lower bound establishes fundamental limits on the degree of compression these techniques can achieve without sacrificing computational accuracy.

**Upper Bounds:** In Appendix C, we provide an explicit neural network construction for 2-AND$_{m,m'}$ and Neural Permutation using only $n = O(\sqrt{m'} \log m')$ neurons; given our lower bounds, this is within a $\sqrt{\log m'}$ factor of optimal. The network uses $O(m' \log^2 m')$ parameters, with an average description length of $O(1)$ bits each, matching the parameter description length lower bound within a $\log m'$ factor. This network computes the function exactly (error-free), uses only $O(1)$ layers, and instances can be chained for sequential computation (e.g., series of 2-AND operations). The construction assumes $O(1)$ input feature sparsity (an assumption also compatible with our lower bounds). To the best of our knowledge, this is the first provably correct algorithm for computing a non-trivial function wholly in superposition.

We also introduce *feature influence*, hinted at in (Chan, 2024) and defined below, which measures how many output features an input affects. Feature influence has a significant impact on what techniques are effective in computing in superposition, and for some functions, determines the ability to compute them in superposition at all. Accounting for varying feature influence makes our 2-AND construction fairly involved. Our algorithm partitions the pairwise AND operations based on influence and applies one of three distinct techniques accordingly.

One of these techniques addresses the low-influence case (inputs all have feature influence $\leq m'^{1/4}$), which reflects most real-world scenarios. This technique routes inputs to dedicated, superposed "computational channels" associated with specific outputs, and then uses these channels for computation. This seems foundational to superposition and may be of general interest. In fact, subsequent work by Adler et al. (Adler et al., 2025) demonstrates that this technique emerges naturally in toy networks trained via standard gradient descent. We thus believe this theoretical construct and its analysis capture a phenomenon that may appear in real networks and thus can inform mechanistic interpretability efforts aimed at understanding learned network behaviors.

Finally, we demonstrate generalizations (Appendix G): the $O(1)$ feature sparsity assumption can be extended to any sparsity $v$ with an additional complexity factor of $O(v2^v \log m)$; the algorithms can be utilized in multi-layered networks; and they can be modified to handle $k$-way AND functions.

**Algorithm Sketch:** We here provide a brief summary of our main algorithm, demonstrating the upper bound. We assume the multi-layer perceptron model described in Appendix B.2. Our goal is to demonstrate that $n = O(\sqrt{m'} \log m')$ neurons are sufficient for any 2-AND problem. We divide the problem up into three subproblems and these subproblems will be solved using the three algorithms described in Appendices C and F. In all cases, the structure of the matrices used is depicted in Figure 2. The algorithm proceeds as follows:

- Label each input *light* or *heavy* depending on how many outputs it appears in, where light inputs appear in at most $m'^{1/4}$ outputs and heavy inputs appear in more than $m'^{1/4}$ outputs.

- Label each output as *double light*, *double heavy* or *mixed*, dependent on how many light and heavy inputs that output combines.

- Partition the outputs of the 2-AND problem into three subproblems, based on their output labels. Each input is routed to the subproblems it is used in, and thus may appear in one or two subproblems. Otherwise, the subproblems are solved independently.

- Solve the double light outputs subproblem using **Low-Influence-AND** (App. F.1).

- Solve the double heavy outputs subproblem using **High-Influence-AND** (App. F.2).

- Solve the mixed outputs subproblem using **Mixed-Influence-AND** (App. F.3). This algorithm actually requires a further division into two independent subproblems.

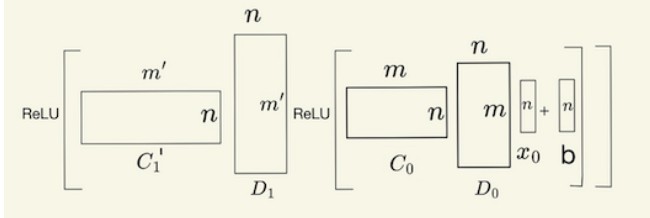

Figure 1: Matrix multiplication structure used by our algorithm.

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

## A  RELATED WORK

Our research builds on the groundbreaking work of Vaintrob, Mendel, and Hänni (Vaintrob et al., 2024), which introduced the algorithmic problem of computing in superposition via a single-layer network for the $k$-AND problem. However, their approach has key limitations that we address. Firstly, their model places only the neurons in superposition, and represents inputs and outputs monosemantically, thereby simplifying the problem and avoiding some of its main challenges (and in fact they show an algorithm for their case that outperforms our lower bound). Our framework requires inputs, neurons, and outputs to be in superposition, which more accurately represents the logical layer of a real network. Secondly, their technique is confined to single-layer networks due to error accumulation, whereas our method eliminates this error, enabling arbitrary network depth. Thirdly, unlike (Vaintrob et al., 2024), we establish for the first time lower bounds on the complexity of computing in superposition.

Subsequent work by Vaintrob, Mendel, Hänni and Chan (Hänni et al., 2024), concurrent with ours, extends their results to compute 2-AND with inputs in superposition using polynomially many layers. However, their result yields the more readily attainable $n = \Theta(m'^{2/3})$, plus an unspecified number of log factors, instead of the almost tight $n = O(\sqrt{m'} \log m')$ we achieve. However, they demonstrate an important and elegant result not addressed here: randomly initialized neural networks are likely to emulate their construction, suggesting that constructions like theirs and ours may occur "in the wild." This work also does not address lower bounds.

Another paper that studies the impact of superposition on neural network computation is (Scherlis et al., 2023). However, they look at a very different question from us: the problem of allocating the capacity afforded by superposition to each feature in order to minimize a loss function, which becomes a constrained optimization problem.

There is a close connection between computation in neural networks and the study of Boolean circuits, particularly circuits with threshold gates (Maass, 1997; Parberry, 1994; Siu et al., 1995), and improvements to our upper bound results would have implications to that field of study. Specifically,

as pointed out in (Williams, 2025), and building on work in (Williams, 2018; 2024), computing 2-AND in a single layer with even a small amount of superposition ($n = o(m'/\log m')$ would suffice) without a feature sparsity assumption would refute the Orthogonal Vectors Conjecture (OVC).

We also contrast our work with the well-studied *network memorization* problem (Park et al., 2021; Vardi et al., 2022; Yun et al., 2019) (also known as *finite sample expressivity*), where a fixed network $N$ uses parameters $P$ to store $Q$ arbitrary input-output pairs $(x_1, y_1), \ldots, (x_Q, y_Q)$: $\forall i, N(P)(x_i) = y_i$. With some assumptions, this requires $\tilde{\Theta}(\sqrt{Q})$ parameters and neurons, and $\tilde{\Theta}(Q)$ bits of parameter description (Vardi et al., 2022). With other assumptions, the known asymptotics are worse: see the related work section in (Yun et al., 2019) and (Vardi et al., 2022) for pointers. Although memorization could represent 2-AND by explicitly providing all input-output pairs, this approach is inefficient. Even assuming feature sparsity $v$ requires $Q \geq \binom{m}{v}$, which leads to an upper bound for computing in superposition of $\tilde{O}(m^{v/2})$ on neurons and depth, and $\tilde{O}(m^v)$ bits of parameter description. This is substantially worse than our result even for the minimal $v = 2$. Furthermore, 2-AND is not able to represent an arbitrary input-output relationship and so memorization lower bounds do not directly apply to 2-AND. Existing memorization lower bounds also rely on VC dimension techniques, requiring assumptions on activation functions and network structure, unlike our information-theoretic lower bounds which are assumption-free in this regard.

We also mention that there is a large body of work (e.g., (Grohs et al., 2024; Hertrich et al., 2021; Vardi & Shamir, 2020)) on lower bounds for the depth, weight size and computational complexity of specific network constructions such as ones with a single hidden layer, or the number of additional neurons needed if one reduces the number of layers of a ReLU neural network (Arora et al., 2018a). In contrast, our work here relates the amount of superposition and the number of parameters in the neural network to the underlying features it computes, a recent discovery in mechanistic interpretability research.

Our work is also inspired by various papers from the research team at Anthropic (Chan, 2024; Elhage et al., 2022b; Olah et al., 2020). Their work influenced our general modeling approach and also inspired our definitions of feature sparsity and feature influence. Their recent work (Ameisen et al., 2025) starts to examine computation as well, using learning techniques to extract the dependencies between features that arise during the computation of trained models on specific inputs.

# B  Modeling Neural Computation

We use two models of computation for the neural network $N(F)$. For lower bounds we work in a general model of *parameter driven* algorithms, which subsumes neural networks (and more). For upper bounds we instantiate this model with a standard neural network architecture.

## B.1  Lower Bound: Parameter Driven Algorithms

In many applications, one fixed network architecture is used across a wide range of tasks, with the choice of task determined only by its parameters. We capture this by the following model. Let $U$ and $V$ be finite sets, and let $\mathcal{F} \subseteq \{ F : U \to V \}$ be a class of functions. We say $T$ is a parameter driven algorithm for $\mathcal{F}$ if there exists a parameterization function $P : \mathcal{F} \to \{0, 1\}^*$ such that

$$\forall F \in \mathcal{F}, \ \forall u \in U, \quad T\big(P(F), u\big) = F(u).$$

Thus $T$ is a single universal architecture that, given parameters $p = P(F)$ and an input $u \in U$, outputs $F(u)$ for any $F \in \mathcal{F}$. In Section D, we generalize this definition to allow $T$ to err on a fraction of inputs.

Because we use this model to prove lower bounds on parameter length, we impose no constraints on how $T$ computes: $T$ may be any function (not necessarily computable) from $\{0, 1\}^* \times U$ to $V$. But if $\mathcal{F}$ is large, then the parameter string must be long—both in the always-correct setting (straightforward) and in the setting where $T$ may make mistakes on some inputs $u \in U$ (requiring a more careful argument). Consequently, our lower bounds apply to neural networks with arbitrary structure and activation functions, and even to non-neural parametrized models.

After establishing parameter lower bounds, we specialize to a network structure using the same kind of square $n \times n$ matrices as in our upper bound model below. For this structure, any lower bound of $B$ parameters implies a lower bound of $\Omega(\sqrt{B})$ neurons.

Parameter driven algorithms are important due to a shift in software usage. Traditional software is often used for databases and analysis, where algorithm descriptions are typically small relative to input size and complexity is dominated by the input. Deep learning changes this balance: the description size (parameters) can be large relative to the input, and complexity is often driven by this size.

### B.2  UPPER BOUND: MULTI-LAYER PERCEPTRONS

For our upper bounds, we restrict to parameter driven algorithms computed by a *multi-layer perceptron* (MLP) of depth $d$ and fixed width $n$. Let $\mathbf{x} \in \{0,1\}^n$ be the input. Each layer $L_i$ applies an affine map followed by a Rectified Linear Unit (ReLU):

$$L_i(\mathbf{z}) \;=\; \mathrm{ReLU}\big(A_i\mathbf{z} + \mathbf{b}_i\big),$$

where $A_i \in \mathbb{R}^{n \times n}$ and $\mathbf{b}_i \in \mathbb{R}^n$ are the parameters of layer $i$. We view each coordinate as one neuron; the $j$th neuron in layer $i$ outputs

$$\max\Big\{0, \sum_{k=1}^{n}[A_i]_{j,k}\, z_k \;+\; [\mathbf{b}_i]_j\Big\}.$$

A depth-$d$ MLP with layers $L_1, \ldots, L_d$ computes

$$N(\mathbf{x}) \;=\; L_d\big(L_{d-1}(\cdots L_1(\mathbf{x})\cdots)\big).$$

The network parameters are $\{A_i, \mathbf{b}_i\}$ for $1 \le i \le d$. We allow $d$ to be arbitrary, though typically $d \ll n$ in practice.

Note that $n$ (the input dimension) need not equal $m$ (the number of Boolean input variables) or $m'$ (the number of Boolean formulas being computed). Instead, the $n$-vector input encodes the $m$ variables under feature sparsity $v$. We say a *layer* computes in superposition if $n < m'$, and an MLP computes in superposition if every layer does. If the input is not initially presented in superposition, we can prepend a transformation that does so.

We omit other common operations (e.g. batch normalization, pooling, or the quadratic activations of (Vaintrob et al., 2024)) and focus on ReLU. Since our lower bounds hold for general activations and our upper bounds nearly match them, such modifications cannot yield much benefit for the problems studied here. It remains open whether relaxing the ReLU restriction can yield significant asymptotic improvements for other problems.

As noted above, feature influence strongly affects network design. Let $F: \{0,1\}^m \to \{0,1\}^{m'}$ have output features $f_1, \ldots, f_{m'}$. For an input variable $x_i$, define its *feature influence* as the number of output features $f_j$ for which there exists a partial assignment $\mathbf{s} \in \{0,1\}^{m-1}$ to the other $(m-1)$ bits such that flipping $x_i$ (with $\mathbf{s}$ fixed) changes $f_j$. Formally, $\mathrm{Infl}(x_i) = \big|\{j \,\exists \mathbf{s} \in \{0,1\}^{m-1} \text{ such that } f_j(\mathbf{s}, x_i = 0) \neq f_j(\mathbf{s}, x_i = 1)\}\big|$. The *maximum*, *average*, and *minimum* feature influences of $F$ are the maximum, average, and minimum of $\mathrm{Infl}(x_i)$ over all input variables $x_i$. Our upper bound algorithms for 2-AND and related problems apply for all feature-influence regimes, though the influence pattern materially affects which techniques the algorithms use.

We do not analyze the complexity of translating $F$ into $N(F)$. However, all algorithms we provide run in time polynomial in $m$, and are likely far more efficient than using traditional training to construct $N(F)$.

## C  UPPER BOUNDS

We give an explicit construction that computes any instance of the Neural Permutation problem and any instance of 2-AND$_{m,m'}$ *in superposition* using only $n = O(\sqrt{m'}\log m')$ neurons and a constant number of $n \times n$ layers. The total number of parameters is $O(m'\log^2 m')$. In this Section and in Appendix F, we assume inputs have at most two active (True) Boolean variables; we show how to generalize this to $v$ active inputs (with a factor of $v2^v\log m$ increase in $n$) in Appendix G.2. An event holds with high probability (w.h.p.) if it occurs with probability at least $1 - m^{-\alpha}$ for an arbitrary constant $\alpha > 0$ (by adjusting constant factors).

**Construction vs. inference.** Our networks are assembled by constructing and then multiplying together larger rectangular matrices during a one-time *construction* phase to produce the $n \times n$ matrices actually used at *inference* time. This lets us construct analyze clean, problem-structured $m$- or $m'$-dimensional operators while ensuring standard MLP inference cost.

## C.1 Input encoding via random compression

Let $\mathbf{y} \in \{0,1\}^m$ denote the monosemantic input. We choose a *compression* matrix $C \in \{0,1\}^{n \times m}$ with i.i.d. Bernoulli($p$) entries where $n = \Theta(\sqrt{m} \log m)$ and $p = \Theta(\log m / n)$. The superposed input is $\mathbf{x} = C\mathbf{y} \in \mathbb{Z}_{\geq 0}^n$.

To argue that no information is lost, define the *decompression* matrix $D \in \mathbb{R}^{m \times n}$ by normalizing each row of $C^\top$: if row $i$ has $r_i$ ones then $D(i,j) = 1/r_i$ when $C^\top(i,j) = 1$ and 0 otherwise. Let $R := DC$. Then $R(i,i) = 1$, and standard Chernoff bounds show that off-diagonals satisfy $R(i,j) = O(1/\log m)$ w.h.p. Hence $\hat{\mathbf{y}} := D\mathbf{x} = R\mathbf{y}$ equals $\mathbf{y}$ on active coordinates and remains $O(1/\log m)$ elsewhere, so thresholding $\hat{\mathbf{y}}$ at $1/2$ recovers $\mathbf{y}$ exactly. This compression/decompression pattern is central to all our constructions.

## C.2 Neural Permutation in superposition

Let $P \in \{0,1\}^{m \times m}$ be a permutation matrix and $\mathbf{x} = C\mathbf{y}$ the compressed input. We want $\mathbf{x}' = C(P\mathbf{y})$ using only $n$-dimensional operators. Consider $T := CPD \in \mathbb{R}^{n \times n}$, $T\mathbf{x} = CPD(C\mathbf{y}) = C(P\mathbf{y}) + C(P\boldsymbol{\epsilon})$, where $\boldsymbol{\epsilon} := (DC - I_m)\mathbf{y}$. As we will see below in our 2-AND analysis, each coordinate of $C(P\boldsymbol{\epsilon})$ is $O(m \log^2 m / n^2)$ w.h.p.; choosing $n = \Theta(\sqrt{m} \log m)$ makes this noise $< \frac{1}{4}$ per coordinate. A constant number of bias+ReLU steps collapse sub-$\frac{1}{2}$ values to 0 and values near 1 to 1, yielding $\mathbf{x}'$ exactly. Thus the permutation can be computed in superposition with $n = O(\sqrt{m} \log m)$. The derivation is similar to the proof of Theorem F.1, in Appendix F.1. *Tightness.* Our lower bounds imply $n = o(\sqrt{m \log m})$ is impossible for Neural Permutation; the residual noise term above is precisely the obstruction.

## C.3 A warm-up special case: single-use 2-AND

We first solve a special case of 2-AND where each input bit appears in at most one output AND gate ("maximum feature influence 1"), hence $m' \leq m/2$. Let $x_0 = Cy_0$ be the compressed input. Using only linear maps in $\mathbb{R}^n$ and coordinatewise ReLUs, the computation is

$$x_1 = \mathrm{ReLU}(C_1' D_1 \, \mathrm{ReLU}(C_0 D_0 x_0 + b))$$

with the following pieces:

- $D_0$ is the decompressor for $x_0$; $\hat{y}_0 = D_0 x_0 \approx y_0$ exposes input bits.
- *Output channels.* Choose $m'$ i.i.d. column specifications $s_1, \ldots, s_{m'} \in \{0,1\}^n$ (Bernoulli($p$) entrywise with $p = \Theta(\log m / n)$). For each output $f_i = y_0(j_i) \wedge y_0(k_i)$, set columns $j_i$ and $k_i$ of $C_0$ equal to $s_i$. Unused inputs get zero columns. Let $b = -\mathbf{1}$.
- $D_1$ "averages over" the support of $s_i$: $D_1(i,j) = 1/|s_i|_1$ if $s_i(j) = 1$, else 0.
- $C_1'$ is a fresh Bernoulli($p$) compressor in $\{0,1\}^{n \times m'}$.

Intuition (Fig. 2, top): $C_0 \hat{y}_0 + b$ maps each intended AND pair to $\{2,1,0\} - 1 \in \{1,0,-1\}$ along the coordinates where $s_i = 1$; the first ReLU keeps only the 1 pattern when $\wedge$ is true. Overlaps between distinct $s_i$ introduce weak activations; $D_1$ spreads and attenuates this noise, and the final $C_1'$ plus thresholding removes it. During construction we explicitly form the $n \times n$ products $C_0 D_0$ and $C_1' D_1$; these are the only matrices used at inference (Fig. 2, bottom). A formal statement and proof for the general low-influence regime appear below, which subsumes this case.

**Input vs. output channels.** In our setup, the column patterning of $C_0$ ties each output $f_i$ to a dedicated code $s_i$, yielding *output channels*: the two inputs feeding $f_i$ are routed into the same $n$-dimensional subspace so that AND reduces to local addition plus thresholding. Effectively, this establishes a dedicated *computational channel* for each output. This technique appears foundational due to its simplicity and potential applicability to more general functions, raising the question of

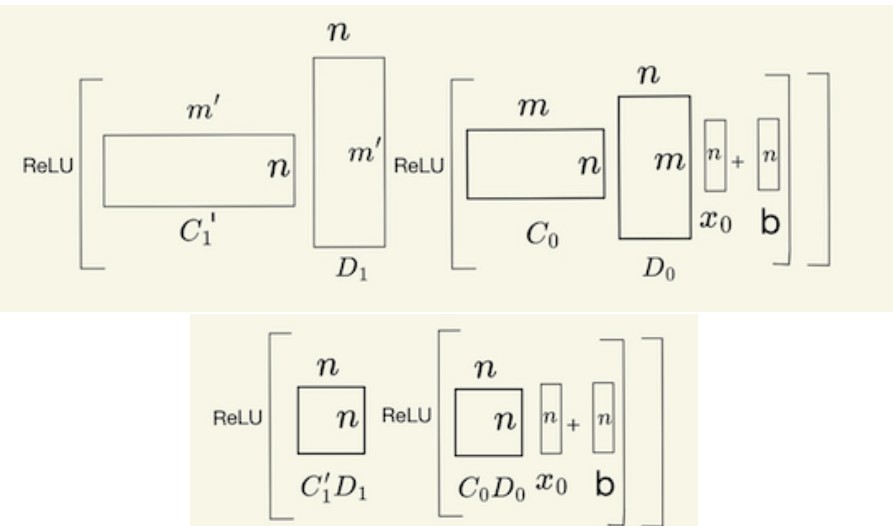

Figure 2: Top: single-use construction via output channels. Bottom: products precomputed for inference.

whether similar mechanisms emerge in conventionally trained neural networks. Subsequent work (Adler et al., 2025) answers this question in the affirmative: this same technique emerges naturally in networks trained via standard gradient descent. We thus believe this technique and its analysis may be of interest to the study of mechanistic interpretability.

Output channels work best when most inputs have low feature influence, since overlaps between different $s_i$ stay small. By contrast, (Vaintrob et al., 2024) uses *input channels*: each input receives a reusable random code whose overlaps support many pairings—ideal for inputs with high feature influence but with more background activity, so the number of such inputs must be $\ll m'$. Using output channels for heavy inputs risks interference; using input channels for light inputs wastes capacity.

### C.4 HIGH LEVEL OUTLINE OF MAIN ALGORITHM

Our goal is to demonstrate that $n = O(\sqrt{m'} \log m')$ neurons are sufficient for any 2-AND problem. We divide the problem up into three subproblems, dependent on feature influence, and these subproblems will be solved using the three algorithms described here below, and in Appendices F.2, and F.3, respectively. In all cases, we use the same structure of matrices as described above, and depicted in Figure 2. We call this structure the *common structure*. Unless otherwise specified, each of the algorithms only changes the specific way that matrices $C_0$ and $D_1$ are defined. Here is a high level description of our algorithm:

- Label each input *light* or *heavy* depending on how many outputs it appears in, where light inputs appear in at most $m'^{1/4}$ outputs and heavy inputs appear in more than $m'^{1/4}$ outputs.
- Label each output as *double light*, *double heavy* or *mixed*, dependent on how many light and heavy inputs that output combines.
- Partition the outputs of the 2-AND problem into three subproblems, based on their output labels. Each input is routed to the subproblems it is used in, and thus may appear in one or two subproblems. Otherwise, the subproblems are solved independently.
- Solve the double light outputs subproblem using algorithm **Low-Influence-AND** (App. F.1).
- Solve the double heavy outputs subproblem using algorithm **High-Influence-AND** (App. F.2).
- Solve the mixed outputs subproblem using algorithm **Mixed-Influence-AND** (App. F.3). This algorithm actually requires a further division into two independent subproblems.

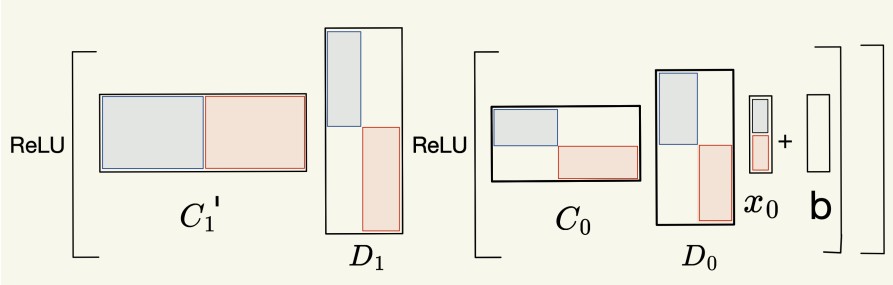

Figure 3: A partition of 2-AND into two subproblems. The red regions compute one subproblem, and the blue regions the other. All entries in other regions will be 0. Note that in $C_1'$, the rows do overlap. This is to set up the outputs of this layer as the inputs to next layer, specifically to allow the same resulting input to appear in up to two subproblems.

To route inputs to the correct subproblems, we use the matrix $C_1'$ of the previous layer, or if this is the first layer, we can insert a preliminary decompress-compress pair prior to $x_0$, followed by a thresholding operation to remove any resulting noise before starting the algorithm above. The partition of the outputs and the computation allocates unique rows and columns to each of the subproblems in every matrix of the computation except $C_1'$ (since that is used to set up the partition for the input to the next layer). As a result, the subproblems do not interfere with each other, and we can treat each subproblem independently. This is depicted in Figure 3 for the case of two subproblems.

In App F.1, we provide a description and proof of correctness for Algorithm **Low-Influence-AND**. Algorithm **High-Influence-AND** (App. F.2) is similar to this, but with its own nuance. The mixed input case is particularly challenging (App. F.3), and requires us to further subdivide the problem into two parts, depending on an even higher threshold of the feature influence of the heavy inputs. It also requires a dedicated **detect-two-active-heavies** gate to prevent catastrophic interference.

We prove in App. F that $n = O(\sqrt{m'} \log m')$ neurons are sufficient for each of the subproblems. Since we only have 3 subproblems, that bound also applies to the overall problem . The challenge in these proofs is bounding the noise incurred as a result of the superposition inherent to the problem. In fact, controlling this noise is the main reason for the additional decompress / compress layer that is added after the output channel encoding our construction performs. Perhaps surprisingly, if this is done the right way, it controls noise instead of adding to it (App. F.1 for details). Our entire construction only requires $O(1)$ layers.

We also demonstrate a number of extensions and further analysis of this protocol in Appendix G. In App. G.1, we demonstrate how to ensure the algorithm can be constructed using an average of $O(1)$ bits per parameter. In App. G.2, we demonstrate how to extend this algorithm to $v$ active inputs, for any $v \le m$, but the resulting $n$ has an exponential dependence on $v$. And in App. G.4 we discuss how to extend this protocol to computing $k$-wise ANDs.

## D  LOWER BOUNDS

We here describe our lower bounds on neurons and parameters for parameter driven algorithms. All of our proofs appear in the Appendix, which also restates the theorems and corollaries stated here. Our first theorem is for the case of parameter driven algorithms that do not make errors. For this case, the result can be shown from a simple counting argument, although in the Appendix we provide a more formal proof of this as a setup for the proof for algorithms that are allowed to make errors. Both our upper and lower bound models assume that a problem instance defines the order of the outputs; this is central to our lower bound proofs (but not required for the upper bound). We also demonstrate, in App. E.3, that the techniques of this section can be extended to the unordered case as well.

**Theorem D.1.** *Let $U$ and $V$ be finite sets, and let $\mathcal{F} \subseteq \{ F : U \to V \}$ be a set of distinct functions. Suppose $T$ is a parameter driven algorithm for $\mathcal{F}$, with parameter function $P(F)$ mapping each $F \in \mathcal{F}$ to a bit string. If $T\big(P(F), u\big) = F(u)$ for all $F \in \mathcal{F}$ and all $u \in U$ then for almost all $F \in \mathcal{F}$, we have $\big|P(F)\big| \ge \log_2\big|\mathcal{F}\big|$.*

We now extend this to allow a parameter driven algorithm to make mistakes on some inputs, as tyically happens in real neural networks. We consider two types of errors:

**Probabilistic errors:** The algorithm's execution can include random sampling (e.g., randomized choices in the neural network), so that for each input $x$, the output may be incorrect with some probability $< \frac{1}{2}$. In this case, we can sample the output of $(T(P(F), x)$ multiple times per input. This yields the same lower bound as the error free case, and so we do not consider this scenario further.

**Systematic errors on a subset of inputs:** Instead, there may be a subset of the possible inputs $u$ on which $T(P(F), u)$ permanently disagrees with $F$. Specifically, for any $F$ in a family $\mathcal{F}$, and some $\epsilon < 0.5$, $T$ is correct for at least a $(1 - \epsilon)$-fraction of $u \in U$ (but possibly wrong on the rest). We say $T$ $\epsilon$-correctly computes $\mathcal{F}$ if for every $F \in \mathcal{F}$, there is a subset $U_F \subseteq U$ with $\frac{|U_F|}{|U|} \geq (1 - \epsilon)$ such that $T(P(F), u) = F(u)$ for all $u \in U_F$.

We cannot hope for as strong a lower bound with these kinds of errors as for the error-free case. Consider for example, a class of functions $\mathcal{F}$ that only differ on a single input $\bar{u} \in U$: $\forall F_1, F_2 \in \mathcal{F}$, $\forall u \in U - \{\bar{u}\}$, $F_1(u) = F_2(u)$. In this case $T$ can always return the same (incorrect) value on $\bar{u}$ and return the correct value on all other inputs. This $T$ requires no parameters, despite always being correct except for a single input.

Instead, we focus on a subset of functions in $\mathcal{F}$ that can always be distinguished from each other. Specifically, we say that $\mathcal{F}' \subseteq \mathcal{F}$ is $\beta$-robust if for all $F_1, F_2 \in \mathcal{F}'$ with $F_1 \neq F_2$, there exists $U'_{F_1 F_2} \subseteq U$ such that $\frac{|U'_{F_1 F_2}|}{|U|} > \beta$ and $\forall u \in U'_{F_1 F_2}$, $F_1(u) \neq F_2(u)$. In other words, $F_1$ and $F_2$ map strictly more than a fraction of $\beta$ of the inputs to different outputs. We use a $\beta$-robust $\mathcal{F}'$ as an error correcting code with Hamming distance $\beta|U|$, where every $F \in \mathcal{F}'$ is a codeword with every $u \in U$ providing one symbol $F(u)$ for that codeword.

**Theorem D.2.** *Let $\epsilon < 0.5$, and suppose $\mathcal{F} \subseteq \{F : U \to V\}$ contains a non-empty $\beta$-robust subset $\mathcal{F}' \subseteq \mathcal{F}$ with $\beta \geq 2\epsilon$. Let $T$ be any parameter driven algorithm that $\epsilon$-correctly computes $\mathcal{F}$. For each $F \in \mathcal{F}$, let $P(F)$ be its parameter description. For almost all $F \in \mathcal{F}'$, $|P(F)| \geq \log |\mathcal{F}'|$.*

**Corollary D.2.1.** *Let $\epsilon < 0.5$, and suppose $T$ $\epsilon$-correctly computes the Neural Permuation problem. Any such $T$ requires a parameter description of length at least $\log[((1 - 2\epsilon)|U|)!] = \Omega(|U| \log |U|)$.*

**Corollary D.2.2.** *For any $m$, $m'$ with $m' \leq \binom{m}{2}$, let $T$ be any parameter driven algorithm that computes 2-$AND_{m,m'}$ $\epsilon$-correctly. $T$ requires a parameter description length of at least $\Omega(m' \log m')$.*

Note that we have made no assumptions here about whether the inputs and/or outputs are stored in superposition, and so this bound applies in all four combinations of superposition or not. Also, note that we can assume that $m' \geq \frac{m}{2}$ since if $m'$ is smaller than that, then we can remove any unused input entries from the problem, thereby reducing $m$. Finally, we again point out that for any neural network in our upper bound model (and thus using square matrices) and a constant number of bits per parameter, these lower bounds imply that the number of neurons required is $\Omega\left(\sqrt{m' \log m'}\right)$. In the Appendix, we also provide a possible way to extend our lower bound techniques to large language models.

# E  FULL PROOFS OF LOWER BOUNDS

We here provide a full version of our lower bound section. Note that we repeat all of the material from Section D, but also provide proof details. In Section E.2 we also provide a discussion on how it might be possible to extend these results to large language models. In Section E.3, we show how to extend our results to the case where the ordering of outputs is left unspecified. We start by assuming that the parameter driven algorithm does not make any errors but will add errors to the mix later below. For the error-free case, we are much more formal than is necessary; we do so in order to set up a framework that makes it much easier to demonstrate the lower bound for when the parameter driven algorithm can make mistakes.

**Theorem E.1.** *Let $U$ and $V$ be finite sets, and let $\mathcal{F} \subseteq \{F : U \to V\}$ be a set of distinct functions. Suppose $T$ is a parameter driven algorithm for $\mathcal{F}$, with parameter function $P(F)$ mapping each $F \in \mathcal{F}$ to a bit string. If*
$$T(P(F), u) = F(u) \quad \text{for all } F \in \mathcal{F} \text{ and all } u \in U,$$

*then for* almost all $F \in \mathcal{F}$, *we have* $\big|P(F)\big| \geq \log_2 |\mathcal{F}|$.

*Proof.* Suppose, for the sake of contradiction, that there exist *many* functions $F \in \mathcal{F}$ whose parameters $\big|P(F)\big|$ are strictly less than $\log_2 |\mathcal{F}|$. We will show how this leads to a communication protocol that transmits $|\mathcal{F}|$ distinct messages using fewer than $\log_2 |\mathcal{F}|$ bits for many of those messages, contradicting basic principles of information theory (e.g., via Kolmogorov complexity).

For simplicity, we assume that $|\mathcal{F}|$ is a power of 2, but this technique generalizes to arbitrary finite $|\mathcal{F}|$. Denote by $B$ a bijection $B : \mathcal{F} \to \{0, 1\}^k$. Consider two parties, Alice and Bob:

**Setup:**

- Both Alice and Bob know the algorithm $T$, the function class $\mathcal{F}$, and the parameters $P(F)$ for each $F \in \mathcal{F}$.

- They also agree on the bijection $B$.

**Protocol:**

- Alice receives a $k$-bit string $s$.

- Alice looks up $F = B^{-1}(s) \in \mathcal{F}$.

- Alice sends Bob the string $P(F)$. By our assumption, $\big|P(F)\big| < k$ for many $F$.

- Bob computes $T\big(P(F), u\big)$ for all $u \in U$. Because the functions in $\mathcal{F}$ are distinct and $T$ agrees with $F$ on all $u \in U$, Bob can uniquely identify $F$.

- From $F$, Bob recovers $s = B(F)$.

Because Bob can recover $s$ from fewer than $k$ bits, we have compressed $k$-bit messages into fewer than $k$ bits for *many* possible messages—contradicting the fact that you cannot reliably encode all $k$-bit messages into fewer than $k$ bits. Hence only a negligible fraction of the functions in $\mathcal{F}$ can have $\big|P(F)\big| < k$, establishing that $\big|P(F)\big| \geq \log_2 |\mathcal{F}|$ for almost all $F$. $\qquad\square$

Note that this theorem does not claim a parameter-length lower bound for any *particular* function $F \in \mathcal{F}$. Rather, it asserts that if you want a *single* network (or any single "universal" structure) to compute *all* functions in $\mathcal{F}$ on inputs in $U$, then for the vast majority of those functions, the parameter description must be at least $\log_2 |\mathcal{F}|$ bits. This parallels the usual Kolmogorov complexity result: almost all objects in a large set require long descriptions, though specific individual objects can sometimes be described more succinctly.

E.1 PARAMETER DRIVEN ALGORITHMS WITH ERRORS

We now extend the previous framework to allow a parameter driven algorithm to make mistakes on some inputs, a scenario that arises in real neural networks. We consider two ways that errors could arise:

1. **Probabilistic errors.**

   The algorithm's execution can include random sampling (e.g., randomized choices in the neural network), so that for each input $x$, the output may be incorrect with some probability. In communication terms, Bob could simply sample the output of $T\big(P(F), x\big)$ multiple times per input. If the probability of a correct output exceeds $\frac{1}{2}$, then with high confidence Bob can recover the correct behavior of $F$. This yields the same contradiction as before, and we do not consider this scenario further.

2. **Systematic errors on a subset of inputs.**

   Instead, there may be a subset of the possible inputs $u$ on which $T\big(P(F), u\big)$ permanently disagrees with $F$. Specifically, for any $F$ in a family $\mathcal{F}$, and some $\epsilon < 0.5$, $T$ is correct for at least a $(1 - \epsilon)$-fraction of $u \in U$ (but possibly wrong on the rest). We say $T$ $\epsilon$-*correctly computes* $\mathcal{F}$ if for every $F \in \mathcal{F}$, there is a subset $U_F \subseteq U$ with $\frac{|U_F|}{|U|} \geq 1 - \epsilon$ such that

   $$T\big(P(F), u\big) \;=\; F(u) \quad \text{for all} \quad u \in U_F.$$

We cannot hope for as strong a lower bound in the presence of these kinds of errors as we did for the error-free case. Consider for example, a class of functions $\mathcal{F}$ where all the functions only differ on a single input $\bar{u} \in U$: if $\forall F_1, F_2 \in \mathcal{F}$, $u \in U - \{\bar{u}\}$, $F_1(u) = F_2(u)$, $\bar{u}$ is always miscomputed by $T$ to the same value, and all other inputs are always computed correctly, then no parameters are needed to distinguish among those functions, despite $T$ only being incorrect on a single input.

Instead, we focus on a subset of functions in $\mathcal{F}$ that can always be distinguished from each other. Specifically, we say that $\mathcal{F}' \subseteq \mathcal{F}$ is $\beta$-robust if for all $F_1, F_2 \in \mathcal{F}'$ with $F_1 \neq F_2$, there exists $U'_{F_1 F_2} \subseteq U$ such that $\frac{|U'_{F_1 F_2}|}{|U|} > \beta$ and $\forall u \in U'_{F_1 F_2}$, $F_1(u) \neq F_2(u)$. In other words, $F_1$ and $F_2$ map strictly more than a fraction of $\beta$ of the inputs to different outputs. We will use a $\beta$-robust $\mathcal{F}'$ in our protocol as an error correcting code with Hamming distance $\beta|U|$, where every $F \in \mathcal{F}'$ is a codeword with every $u \in U$ providing one symbol $F(u)$ for that codeword.

Let $\epsilon < 0.5$, and suppose $\mathcal{F} \subseteq \{\, F : U \to V \,\}$ contains a *non-empty* $\beta$-robust subset $\mathcal{F}' \subseteq \mathcal{F}$ with $\beta \geq 2\epsilon$. Let $T$ be any parameter driven algorithm that $\epsilon$-correctly computes $\mathcal{F}$. For each $F \in \mathcal{F}$, let $P(F)$ be its parameter description.

**Theorem E.2.** *For almost all $F \in \mathcal{F}'$, $|P(F)| \geq \log |\mathcal{F}'|$.*

*Proof.* As before, we prove this by constructing a communication protocol that would represent $|\mathcal{F}'|$ messages into fewer than $\log_2 |\mathcal{F}'|$ bits, contradicting standard information-theoretic limits.

**Setup:**

- Alice and Bob are both given $T$, $\epsilon$, $\mathcal{F}$, $\mathcal{F}'$, and $P(F)$ for all $F \in \mathcal{F}$.

- Alice and Bob also agree on a bijection $B$ from $\mathcal{F}'$ to $\{0,1\}^{\log_2 |\mathcal{F}'|}$.

**Protocol:**

- Alice is given a message $s$, a $\log_2 |\mathcal{F}'|$-bit string.

- Alice identifies $F = B^{-1}(s) \in \mathcal{F}'$.

- Alice transmits the parameter string $P(F)$ to Bob.

- Bob uses $T\big(P(F), u\big)$ for all $u \in U$ to define a function $F^* : U \to V$.

- Because $T$ is $\epsilon$-correct on $\mathcal{F}$ (and hence on $\mathcal{F}' \subseteq \mathcal{F}$), $F^*$ agrees with $F$ on at least $(1-\epsilon)|U|$ inputs.

- Given the $\beta$-robustness of $\mathcal{F}'$ (with $\beta \geq 2\epsilon$), no other $F' \neq F$ in $\mathcal{F}'$ can match $F^*$ on as many inputs. Hence Bob can recover $F$ by picking the function in $\mathcal{F}'$ closest to $F^*$.

- Finally, Bob determines that $s = B(F)$.

If too many functions $F \in \mathcal{F}'$ had short parameter encodings $\big|P(F)\big| < \log_2 |\mathcal{F}'|$, Alice and Bob would transmit $\log_2 |\mathcal{F}'|$-bit messages in fewer than $\log_2 |\mathcal{F}'|$ bits—an impossibility by standard information-theoretic arguments (e.g., Kolmogorov complexity). Therefore, for almost all $F \in \mathcal{F}'$, the parameter length must satisfy $|P(F)| \geq \log_2|\mathcal{F}'|$. $\qquad \square$

We next demonstrate how to apply this to the 2-AND function. As an intermediate step, we first prove a lower bound on parameter driven algorithms for the Neural Permutation problem, where $\mathcal{F}$ is the class of all permutations on a set $U$. Let $\epsilon < 0.5$, and suppose $T$ $\epsilon$-correctly computes each permutation $F \in \mathcal{F}$.

**Corollary E.2.1.** *Any such $T$ requires a parameter description of length at least* $\log[((1 - 2\epsilon)|U|)!] = \Omega(|U| \log |U|)$.

*Proof.* By Theorem E.2, it suffices to exhibit a $\beta$-robust subset $\mathcal{F}' \subseteq \mathcal{F}$ of size $((1 - 2\epsilon)|U|)!$ with $\beta \geq 2\epsilon$. We construct $\mathcal{F}'$ greedily: pick any unused permutation $F$, add it to $\mathcal{F}'$, then remove from consideration all permutations that do not differ from $F$ on at least $(2\epsilon)|U|$ inputs. Each chosen permutation eliminates at most $\binom{|U|}{2\epsilon|U|} (2\epsilon|U|)!$ permutations, so we can place at least

$$\frac{|U|!}{\binom{|U|}{2\epsilon|U|} (2\epsilon|U|)!} = ((1 - 2\epsilon)|U|)!$$

permutations into $\mathcal{F}'$. These permutations differ from each other on more than a fraction $2\epsilon$ of inputs, as desired. Note that the subset we have constructed is essentially a permutation code (Slepian, 1965). $\qquad\square$

**Corollary E.2.2.** *For any $m$, $m'$ with $m' \leq \binom{m}{2}$, let $T$ be any parameter driven algorithm that computes* 2-AND$_{m,m'}$ *$\epsilon$-correctly. $T$ requires a parameter description length of at least $\Omega(m' \log m')$.*

*Proof.* Fix $m$ and $m' \leq \binom{m}{2}$. We construct a class $\mathcal{F}$ of 2-AND$_{m,m'}$ instances and a set $U$ of inputs which demonstrate this bound. First, choose any set $S \subseteq \{ (i, j) \colon 1 \leq i < j \leq m \}$ of size $|S| = m'$. Each element of $S$ is a pair of input coordinates $(i, j)$. Then consider all $F \colon \{0, 1\}^m \to \{0, 1\}^{m'}$ which compute the ANDs of exactly those pairs in $S$, including all different orderings across the $m'$ output positions. Concretely, for each permutation $\sigma$ of $\{1, 2, \ldots, m'\}$, define

$$F_\sigma(x_1, \ldots, x_m) =$$

$$\left( x_{i_{\sigma(1)}} \wedge x_{j_{\sigma(1)}},\ x_{i_{\sigma(2)}} \wedge x_{j_{\sigma(2)}},\ \ldots,\ x_{i_{\sigma(m')}} \wedge x_{j_{\sigma(m')}} \right),$$

where $\{(i_k, j_k)\}_{k=1}^{m'}$ is an enumeration of the pairs in $S$. Let $\mathcal{F} = \{ F_\sigma \mid \sigma$ is a permutation of $\{1, \ldots, m'\}\}$ and thus $|\mathcal{F}| = m'!$.

Note that if $m' \ll \binom{m}{2}$, then most two-hot inputs will have all entries of the output evaluate to 0; hence if $U$ were to consist of all two-hot inputs, $T$ could compute $\epsilon$-correctly by simply producing the all 0s result for every input. Instead, we restrict $U$ to a specific set of $m'$ *two-hot inputs*, one for each pair in $S$. Concretely, for each $(i_k, j_k) \in S$, define $u_k = e_{i_k} + e_{j_k} \in \{0, 1\}^m$, where $e_r$ is the standard basis vector with a 1 in position $r$ and 0 in every other position. Thus each $u_k$ has exactly two coordinates equal to 1. Let $U = \{u_1, \ldots, u_{m'}\}$, and so $|U| = m'$. Each $F_\sigma \in \mathcal{F}$ induces a *distinct* labelling of the inputs $U$ according to the permutation $\sigma$. The Corollary now follows from the exact same argument as was used for the permutation function. $\qquad\square$

Note that we have made no assumptions here about whether the inputs and/or outputs are stored in superposition, and so this bound applies in all four combinations of superposition or not. Also, note that we can assume that $m' \geq \frac{m}{2}$ since if $m'$ is smaller than that, then we can remove any unused input entries from the problem, thereby reducing $m$. Finally, we again point out that for any neural network in our upper bound model (and thus using square matrices) and a constant number of bits per parameter, this lower bound implies that the number of neurons required is $\Omega\left(\sqrt{m' \log m'}\right)$.

## E.2 POSSIBLE EXTENSIONS TO LLM PARAMETERIZATION

Although one might argue that a single trained large language model (LLM) represents only one function and therefore falls outside our lower bound framework, modern neural architectures are typically designed to implement a vast family of functions. The architecture's high expressibility is realized through its parameters, which in turn are specified by training. Our lower bound applies to this underlying expressibility, prior to training, rather than to a single, fully trained model.

To see how one could potentially establish a parameterization lower bound for LLMs, consider training a network architecture $\Upsilon$ on a corpus $D$, yielding a model $\Upsilon(D)$, which computes a function $F(\Upsilon(D))$ (where different models can still compute the same underlying function). Consider using two very different datasets—e.g., $D_E$, an entirely English text versus $D_M$, an entirely Mandarin text. It seems likely these two training regimes yield significantly different functions: $F(\Upsilon(D_E)) \neq_{2\epsilon} F(\Upsilon(D_M))$, where we use $\neq_{2\epsilon}$ to denote that two functions differ on a fraction of at least $2\epsilon$ of their inputs. More drastically, let $D$ be a corpus of length $r$, measured in total words. Let $D'$ be a random permutation of the entire sequence of words. With high probability, $D'$ disrupts most of the natural linguistic structure in $D$, and so it seems likely $F(\Upsilon(D')) \neq_{2\epsilon} F(\Upsilon(D))$.

A stronger claim is that for two distinct random permutations $D''$ and $D'$ of the original corpus $D$, training $\Upsilon$ on each would yield two functions different from each other. Both permutations jumble the original corpus but do differently jumbled training sets lead to functions different from each other? If we could show that for every pair $D', D''$ of sufficiently different permutations of $D$, we have $F(\Upsilon(D'')) \neq_{2\epsilon} F(\Upsilon(D'))$, we could use the techniques above to provide a lower bound of $\Omega(r \log r)$ on the length of the parameter description needed to specify $\Upsilon$. While we do not attempt such a proof here, investigating the size of this permutation-based function family could be a fruitful direction for future work.

### E.3   LOWER BOUND FOR UNORDERED 2-AND

We here adapt the lower bound proof to the case where the problem is to compute a *set* of $m'$ logical ANDs, without regard to the order in which the results appear at the output. This is a more relaxed condition than the one presented in the original problem, but as we will show, it still necessitates a significant number of parameters. An instance of the *Unordered 2-AND$_{m,m'}$* problem is defined by a *set* $G = \{f_1, f_2, ..., f_{m'}\}$, where each $f_i$ is the logical AND of a unique pair of the $m$ input variables.

The crucial difference here is that $G$ is a set, not an ordered tuple. A neural network $N(G)$ correctly solves this problem if, for any input $u$, its output is a multiset corresponding to $\{f_1(u), f_2(u), ..., f_{m'}(u)\}$. The task of the parameter-driven algorithm is to configure the network to compute the correct set of logical operations specified by $G$. The class of all such problems, denoted $\mathcal{G}_{m,m'}$, consists of all possible sets of $m'$ distinct 2-input ANDs chosen from the $m$ variables. The total number of unique pairs of variables is $\binom{m}{2}$. Therefore, the size of this problem class is:

$$|\mathcal{G}_{m,m'}| = \binom{\binom{m}{2}}{m'}$$

This removes the $m'!$ factor present in the ordered version of the problem. We first present a straightforward bound for the error-free case and then provide a more detailed proof for the robust case where the computation is allowed to have up to a constant fraction of errors.

**Error-Free Computation** For a parameter-driven algorithm that computes the unordered 2-AND problem without errors, a simple counting argument suffices.

**Corollary E.2.3.** *Let $T$ be any parameter-driven algorithm that correctly computes every function in the unordered 2-AND$_{m,m'}$ class. For almost all functions $F \in \mathcal{G}_{m,m'}$, the length of the parameter string $P(F)$ must be at least $\log_2 |\mathcal{G}_{m,m'}|$. This implies a lower bound of:*

- $\Omega(m' \log m)$ *bits when $m'$ is polynomially smaller than $\binom{m}{2}$.*

- $\Omega(m')$ *bits when $m'$ is a constant factor smaller than $\binom{m}{2}$.*

*Proof.* The proof directly follows Theorem E.1. Since the algorithm $T$ must be able to distinguish between all $\binom{\binom{m}{2}}{m'}$ possible functions, there must be at least that many distinct parameter settings. By the same information-theoretic argument, for almost any function $F$, its parameter description $P(F)$ must have length at least $\log_2\left(\binom{\binom{m}{2}}{m'}\right)$.

We can bound this quantity using $\left(\frac{N}{k}\right)^k \leq \binom{N}{k}$, where $N = \binom{m}{2}$ and $k = m'$. Taking the logarithm gives:

$$|P(F)| \geq \log_2 \binom{N}{k} \geq k \log_2 \left(\frac{N}{k}\right) = m' \log_2 \left(\frac{\binom{m}{2}}{m'}\right)$$

This expression simplifies to $\Omega(m' \log m)$ or $\Omega(m')$ under the conditions stated in the corollary, establishing the lower bound for the error-free case. $\qquad\square$

**Computation with Errors** For the unordered case of computation with errors, we demonstrate a linear in $m'$ lower bound, provided that the 2AND problem is not too close to complete, and the correctness of the network is slightly higher than was required in the ordered case.

**Corollary E.2.4.** *For any $m, m'$ with $m' \leq \binom{m}{2}/2$, let $T$ be any parameter-driven algorithm that computes unordered 2-AND$_{m,m'}$ $\epsilon$-correctly for any constant $\epsilon < 1/8$. For almost all such problems, $T$ requires a parameter description length of at least $\Omega(m')$.*

*Proof.* We follow the structure dictated by Theorem E.2 by constructing a specific, large, and robust subset of the function class $\mathcal{G}_{m,m'}$.

1. **Construct a Base Set and Input Set.** Assume $m$ is large enough such that we can choose $2m'$ distinct pairwise ANDs (distinct here means every two pairwise ANDs differ on at least one of the two inputs). Let this be our *base set of ANDs*, $A = \{a_1, a_2, \ldots, a_{2m'}\}$. Let $U$ be the set of $2m'$ corresponding two-hot inputs, $U = \{u_1, \ldots, u_{2m'}\}$, where input $u_k$ is constructed to make AND $a_k$ evaluate to 1, while all other ANDs $a_j$ (for $j \neq k$) evaluate to 0. Thus, $|U| = 2m'$.

2. **Construct a Robust Function Collection $\mathcal{F}'$.** We define a collection of functions $\mathcal{F}' \subseteq \mathcal{G}_{m,m'}$. Each function $F \in \mathcal{F}'$ is defined by a set of exactly $m'$ ANDs chosen from the base set $A$. We can represent each such function by a binary string of length $2m'$, where the $k$-th bit is 1 if $a_k \in F$ and 0 otherwise.

We construct $\mathcal{F}'$ to be a collection of such functions where the Hamming distance between the corresponding binary strings of any two functions, $F_1, F_2 \in \mathcal{F}'$, is at least $d = m'/2$. Such a collection is an error-correcting code. By the Gilbert-Varshamov bound, we know there exists such a collection $\mathcal{F}'$ of size:

$$|\mathcal{F}'| \geq \frac{2^{2m'}}{\sum_{i=0}^{d-1} \binom{2m'}{i}} = \frac{2^{2m'}}{\sum_{i=0}^{m'/2-1} \binom{2m'}{i}}$$

Using standard bounds on binomial sums (e.g., from Chernoff bounds), the denominator is at most $2^{2m'} \cdot e^{-m'/8}$. This guarantees the existence of a collection $\mathcal{F}'$ whose size is exponential in $m'$, i.e., $|\mathcal{F}'| \geq 2^{m'/8 \ln 2}$.

3. **Show the Collection is Robust.** We now show that this collection $\mathcal{F}'$ is $\beta$-robust with $\beta = 1/4$ over the input set $U$. Consider any two distinct functions $F_1, F_2 \in \mathcal{F}'$. By construction, their corresponding binary indicator vectors have a Hamming distance of at least $m'/2$. This means their defining sets of ANDs, $S_1$ and $S_2$, have a symmetric difference $|S_1 \Delta S_2| \geq m'/2$.

An input $u_k \in U$ is one where the outputs of $F_1$ and $F_2$ differ if and only if one of the functions contains the AND $a_k$ and the other does not. This is precisely the condition that the $k$-th bit of their indicator vectors differs. Therefore, the number of inputs in $U$ for which $F_1$ and $F_2$ produce different outputs is exactly the Hamming distance between their representations, which is at least $m'/2$.

The fraction of inputs in $U$ on which they differ is therefore at least $\frac{m'/2}{|U|} = \frac{m'/2}{2m'} = \frac{1}{4}$. Thus, the set $\mathcal{F}'$ is $1/4$-robust.

4. **Apply the Lower Bound Theorem.** We have constructed a $1/4$-robust subset $\mathcal{F}'$ of functions of size $2^{\Omega(m')}$. According to Theorem E.2, for a parameter-driven algorithm $T$ to be $\epsilon$-correctly compute these functions, we require the robustness $\beta$ to satisfy $\beta \geq 2\epsilon$. With $\beta = 1/4$, this condition becomes $1/4 \geq 2\epsilon$, or $\epsilon \leq 1/8$.

Therefore, for any algorithm that computes the unordered 2-AND problem with an error rate $\epsilon < 1/8$, it must be able to distinguish between the functions in our robust set $\mathcal{F}'$. By Theorem E.2, for

almost all functions $F \in \mathcal{F}'$, the parameter length must satisfy:

$$|P(F)| \geq \log_2 |\mathcal{F}'| \geq \log_2(2^{m'/8 \ln 2}) = \Omega(m')$$

This establishes a lower bound of $\Omega(m')$ on the number of parameter bits required. $\square$

## F  FURTHER DETAILS AND CORRECTNESS OF OUR 2-AND CONSTRUCTION

We will prove that $n = O(\sqrt{m'} \log m')$ neurons are sufficient for each of the subproblems of our overall procedure described in Appendix C.4 and thus that bound also applies to the overall problem since there are a constant number of subproblems. We note that when some of the outputs are placed in a subproblem, the inputs that remain may go from being heavy to light (since they have lost some of their outputs). We use the convention that we continue to classify such inputs with their original designation. Also, one or two of the subproblems may become much smaller than the original problem. However, when we partition the problem into these subproblems, we will treat each subproblem as being of the same size as the original input: we will use a value of $n = O(\sqrt{m'} \log m')$ for each of the subproblems, regardless of how small it has become.

We also now clarify what we meant above by thresholding the entries of a vector. This is an operation on an $n$-vector that forces all entries to either 0 or 1. This thresholding (mapping values $< 1/2$ to 0 and $\geq 1/2$ to 1) can be implemented using two ReLU layers. First, compute $y = \text{ReLU}(x - 1/4)$ for each entry. Second, compute $1 - ReLU(-2 * y + 1)$ for each entry, which guarantees the objective. We can do this any time we have an intermediate result that is in superposed representation, and so we only need to be concerned with getting our superposed results to be close to correct. Note that we cannot use ReLU when an intermediate result is in its uncompressed form, since that would require ReLU to operate on $m \gg n$ entries.

In the analysis that follows, we frequently make use of Chernoff bounds (Mitzenmacher & Upfal) to prove high probability results. In all cases, we use the following form of the bound:

$$\Pr(X \geq (1 + \delta)\mu) \leq e^{-\frac{\delta^2 \mu}{2 + \delta}}, \quad 0 \leq \delta,$$

As mentioned above, our algorithms are always correct for all inputs. In our method of constructing the algorithm, there is a small probability that the construction will not work correctly (with high probability it will work). However, we can detect whether this happened by trying all pairs of inputs being active, and verifying that the algorithm works correctly. If it does not, then we restart the construction process from scratch, repeating until the algorithm works correctly. These restarts do not add appreciably to the expected running time of the process of constructing the algorithm. Also, the resulting neural network itself never uses randomness. This also means that we can chain together an unlimited number of these constructions for different 2-AND (and other) functions, without accumulating error or probability of an incorrect result.

### F.1  ALGORITHM FOR DOUBLE LIGHT OUTPUTS

We first handle the case where all inputs are light. This means that the maximum feature influence is at most $m'^{1/4}$. We show that in this case $n = O(\sqrt{m'} \log m')$ is sufficient.

Algorithm 1, for double light outputs, called Algorithm **Low-Influence-AND**, is described below. The matrix $C_0$ is depicted in Figure 4. We point out that we are still using output channels here, since we are actively routing inputs that need to be paired up to the channels specified by the column specifications. We then combine all the channel specifications for a given input into a single column for that input. We say that a neural network algorithm *correctly computes in superposition* $x_1$ from $x_0$, if $x_0$ and $x_1$ are represented in superposition, and for any input $x_0$, $x_1$ represents the output of the 2-AND problem for that $x_0$, with all intended 0s being numerically 0 and all intended 1s numerically 1.

**Theorem F.1.** *When the maximum feature influence is at most $m'^{1/4}$, at most 2 inputs are active, and $n = O(\sqrt{m'} \log m')$, Algorithm* **Low-Influence-AND** *with high probability correctly computes in superposition $x_1$ from $x_0$.*

Note that this subsumes the single-use case above.

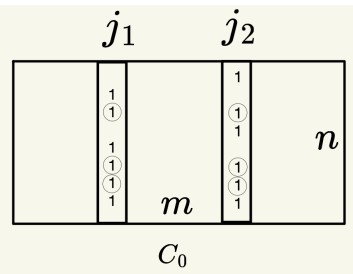

Figure 4: The matrix $C_0$ for **Low-Influence-AND**. The circled 1s are those that correspond to $s_i$, where output $i$ computes $j_1 \wedge j_2$, and thus the 1s in those rows will line up between $j_1$ and $j_2$. Other rows with 1s come from different column specifications, and thus only line up by chance, but when that happens it causes spurious 1s to appear after the second ReLU. When there are at most $O(m'^{1/4} \log m')$ total 1s in each column, it is likely there will be $O(\log m')$ such spurious 1s. However, since $n = O(\sqrt{m'} \log m')$, if there were more 1s in both columns, the number of spurious 1s would become too large to handle. This is why $m'^{1/4}$ represents such an important phase change for what techniques are effective for this problem.

---

**Algorithm 1 Low-Influence-AND** (double-light outputs)

---

1: Require Number of inputs $m$, number of outputs $m'$, dimension $n = \Theta(\sqrt{m'} \log m')$, sparsity $p = \Theta(\log m / n)$.
2: Require Compressed input $x_0 \in \mathbb{Z}_{\geq 0}^n$ with $x_0 = Cy_0$ for some $y_0 \in \{0,1\}^m$; decompressor $D_0$ built from $C$ (so $D_0 C \approx I_m$).
3: Require Output pairs $\{(j_i, k_i)\}_{i=1}^{m'}$ defining $f_i = y_0(j_i) \wedge y_0(k_i)$.
4: Ensure $x_1 \in \mathbb{R}^n$ is a compressed representation of the $m'$ AND outputs (for handoff to the next layer).
5: **Sample output-channel codes:** For each $i \in [m']$, draw a column specification $s_i \in \{0,1\}^n$ with i.i.d. Bernoulli($p$) entries.
6: **Construct $C_0 \in \{0,1\}^{n \times m}$:** For each input $j \in [m]$,

$$C_0(:,j) \leftarrow \begin{cases} \bigvee_{i: j \in \{j_i, k_i\}} s_i & \text{if } j \text{ participates in at least one output,} \\ \mathbf{0} & \text{otherwise,} \end{cases}$$

i.e., the elementwise OR over the $s_i$ for outputs that include $j$.
7: **Define $D_1 \in \mathbb{R}^{m' \times n}$:** For each $i \in [m']$, set the $i$th row to the normalized code,

$$D_1(i, \cdot) \leftarrow \frac{s_i^\top}{\|s_i\|_1}.$$

8: **Fresh compressor and bias:** Sample $C_1' \in \{0,1\}^{n \times m'}$ i.i.d. Bernoulli($p$). Set $b := -\mathbf{1}_n$.
9: **Expose inputs (decompress):** $\hat{y}_0 \leftarrow D_0 x_0$.
10: **Channel preactivation:** $u \leftarrow C_0 \hat{y}_0 + b$.
11: **Gate the AND pattern:** $z \leftarrow \text{ReLU}(u)$.
12: **Average within channels (denoise):** $w \leftarrow D_1 z$.
13: **Compress for next layer:** $x_1 \leftarrow \text{ReLU}(C_1' w)$.
14: **return** $x_1$.
15: *Compact form:* $x_1 = \text{ReLU}\big(C_1' D_1 \text{ReLU}(C_0 D_0 x_0 + b)\big)$.
16: *Inference note:* Only the $n \times n$ products $W_0 := C_0 D_0$ and $W_1 := C_1' D_1$ are materialized at inference; the network applies $x_1 = \text{ReLU}\big(W_1 \text{ReLU}(W_0 x_0 + b)\big)$.

---

*Proof.* We already demonstrated in our discussion of the single use case that we will get values that are 1 in the entries of $x_1$ that were supposed to be 1s; a very similar argument holds here, and so we only need to demonstrate that the inherent noise of the system does not result in too large values in the entries of $x_1$ that are supposed to be 0s. There are two sources of noise in the system:

(a) Multiplying by the decoding matrix $D_0$ is not perfect: there is the potential to have entries of $D_0 x_0$ that are supposed to be zero but are actually nonzero since each row of $D_0$ can have a 1 in the same column as a row that corresponds to an input that's a 1. Or equivalently, each row $i$ of $D_0$ can have a 1 at a location that lines up with a 1 in the row representing the active $x_0$. We need to show that the resulting noise in $D_0 C_0 x_0$ is small enough to be removed by the first ReLU operation.

(b) Multiplying by the decoding matrix $D_1$ is also not perfect for the same reason. There can be overlap between the different output channels. Furthermore, since an input can be used multiple times (but in this case at most $m'^{1/4}$ times) we can also get 1s in the matrix $\text{ReLU}[C_0 D_0 x_0]$ in places outside the correct output channel. Both of these effects lead to noise after multiplying by $D_1$, and potentially after subsequently multiplying by $C_1'$ as well. We also need to show that this noise is small enough to be removed by the second ReLU operation.

We point out that as long as it is small, the noise of type (a) is removed by the first ReLU operation (right after the multiplication by $C_0$), and thus will not contribute to the noise of type (b). Thus, we can analyze the two types of noise independently. We handle noise of type (b) first. Let $y_1 = D_1 \text{ReLU}[C_0 D_0 x_0]$. Our goal is to show that $\text{ReLU}[C_1' y_1]$ only has non-zero entries in the correct places. Let $x_1' = \text{ReLU}[C_0 D_0 x_0]$.

**Claim F.2.** *Any entry of $C_1' y_1$ that does not correspond to a correct 1 of the 2-AND problem has value at most $\epsilon$ due to noise of type (b) with high probability.*

*Proof.* For any matrix $M$, we will refer to entry $(i, j)$ in that matrix as $M(i, j)$, and similarly we will refer to entry $k$ in vector $V$ as $V(k)$. Let $e$ be the index of any entry of $C_1' y_1$ that should not be a 1. We will show that with high probability the value of the entry $C_1' y_1(e)$ is at most $\epsilon$. Due to our ReLU specific operation, we can assume that all non-zero entries of $x_1'$ are at most 1. We refer to the two active inputs as $i$ and $j$, where $i \neq j$ (there is no type (b) noise if there is only one active input). We first consider the expectation of $C_1' y_1(e)$. In the following expression for $E[C_1' y_1(e)]$, we let $a$ range over the entries of $C_1'$ in row $e$ and $b$ range over the columns of $D_1$. For entry $x_1'(b)$ to be a 1, we need $C_0(b, i) = 1$ and $C_0(b, j) = 1$. For this to translate to a non-zero value in its term of the sum for entry $y_1(a)$, we also need $D_1(a, b) = 1$. Since the entries in $D_0$ will be $O(\frac{1}{\log m'})$ with high probability, this gives the following:

$$
\begin{aligned}
E[C_1' y_1(e)] = O\Bigg( \frac{1}{\log m'} \sum_{a \in 1 \dots m'} \sum_{b \in 1 \dots n} \\
\Pr[C_0(b, i) = 1] \\
\cdot \Pr[C_0(b, j) = 1 \mid C_0(b, i)] \\
\cdot \Pr[D_1(a, b) = 1 \mid C_0(b, i), C_0(b, j)] \\
\cdot \Pr[C_1'(e, a) = 1] \Bigg)
\end{aligned}
\tag{1}
$$

Note that $C_1'(c, e)$ is independent of all the other events we are conditioning on, and so we do not need to condition for the probability associated with that event. The other three events are independent for most terms in the sum, but not so for a small fraction of them. $D_1(a, b)$ is independent of $C_0(b, i)$ whenever $i$ is not used in the output for row $a$ of $D_1$, and similarly for $C_0(b, j)$. $C_0(b, i)$ is independent of $C_0(b, j)$, as long as $D_0(d, b) = 0$, where row $d$ of $D_0$ corresponds to the output that is an AND of $i$ and $j$ (since then we know that for all such entries $b$ of row $d$, $s_d(c) = 0$). The entries $y_1(d)$ where $D_0(d, b) > 0$ are supposed to be non-zero, but they can still contribute noise when multiplied by the compression matrix $C_1'$.

Thus, we will evaluate this sum using two cases: where there is some dependence between any pair of the four probabilities, and where there is not. We deal with the latter case first, and in this case $\Pr[C_1'(e, a) = 1] = \Pr[D_1(a, b) = 1] = O(\log m'/n)$. Using a union bound and the fact that no input is used more than $m'^{1/4}$ times, we see that $\Pr[C_0(b, i) = 1] = O(m'^{1/4} \log m'/n)$ and also

$\Pr[C_0(b, j) = 1] = O(m'^{1/4} \log m'/n)$. This tells us that the contribution of the independent terms is at most

$$O\left(\frac{1}{\log m'} nm' \left(\frac{\log m'}{n}\right)^2 \left(\frac{m'^{1/4} \log m'}{n}\right)^2\right) = O(1).$$

For the case where there is dependency between the different events, we first consider what happens when $i$ is used in the output for row $a$ of $D_1$. In this case, we can simply assume that $D_1(d_c) = 1$ always, which means we lose a factor of $\log m'/n = 1/\sqrt{m'}$ in the above equation. However, since $i$ can be used in at most $m'^{1/4}$ outputs, there are now only $m'^{1/4}$ values of $a$ to consider instead of $m'$, so we also lose a factor $m'^{3/4}$. From this we see that the terms of this case do not contribute meaningfully to the value of the sum, and similarly for when $j$ is used in the output for row $a$. For the case where $D_0(d, b) > 0$, where $i$ and $j$ are used in the output for row $d$, we see that all three of the dependent variables will be 1. However, there is only 1 such row $d$, and we also know that with high probability that row will contain $O(\log m')$ 1s. Thus, the number of terms in the sum is reduced by $m'^{3/2}$ and we still have the $\frac{\log m'}{n}$ from $\Pr[C_1'(e, a) = 1]$, and so this case does not contribute significantly to the sum either.

To convert this expectation to a high probability result, we can rearrange the terms of the sum to consider only those rows $d$ of $D_1$ that correspond to columns of $C_1'$ where $C_1'(d, e) = 1$ and those columns of $D_1$ that correspond to entries of $x_1'$ where $x_1'(c) = 1$ incorrectly (i.e. $c$ such that there exist $a$ and $b$ such that both $C_0(c, a) = 1$ and $C_0(c, b) = 1$). The number of non-zero entries in a row of $C_1'$ is $O(m' \log m'/n)$ with high probability (from how $C_1'$ is built).

The number of non-zero entries in $x_1'$ is $O(\log m')$ with high probability. Thus, with high probability, we are summing a total of $O(m' \log^2 m'/n) = O(\sqrt{m'} \log m')$ entries of $D_1$. Each of those entries is either $\Theta(1/\log m')$ or 0 and takes on the non-zero value with probability $\log m'/n$. Thus the expectation of that sum is $O(m' \log^2 m'/n^2) = O(1)$. We can define indicator variables on whether or not each such entry of $D_1$ is non-zero. We can assume these random variables are chosen independently, and their expected sum is $O(\log m')$, and so a standard Chernoff bound then demonstrates that the number of non-zero entries in $D_1$ will be within a constant of its expectation with high probability. Thus, $C_1' y_1(e)$ will be $O(1)$ with high probability. We can make that constant smaller than any $\epsilon$ be increasing the size of $n$ by a constant factor dependent on $\epsilon$. $\qquad \square$

Thus, all noise of type (b) will be removed by the second ReLU operation. We now turn to noise of type (a): there can be incorrect non-zeros in the vector $D_0 x_0$ and we want to make sure that any resulting incorrect non-zero entry in the vector $C_0 D_0 x_0$ has size at most $\epsilon$ and thus will be removed by the first ReLU function.

**Claim F.3.** *The amount of type (a) noise introduced to any entry of $C_0 D_0 x_0$ is at most $\epsilon$ with high probability.*

*Proof.* For any row $e$ of $C_0$, let $C_0^e$ be the set of columns $a$ of $C_0$ such that $C_0(e, a) = 1]$. We first show that for any $e$, with high probability, $|C_0^e| = O(\sqrt{m'})$. This follows from how the columns of $C_0$ are chosen: they are defined by the column specifications $s_1, \ldots, s_{m'}$. Every column specification $s_i$ where $s_i(e) = 1$ contributes at most 2 new columns to $C_0^e$ - one for each input used for output $i$. There are $m'$ column specifications, and the entries are all chosen i.i.d., with probability of a 1 being $1/\sqrt{m'}$, and so a straightforward Chernoff bound shows that with high probability there are at most $O(\sqrt{m'})$ specifications $s_i$ where $s_i(e) = 1$. Thus $|C_0^e| = O(\sqrt{m'})$ with high probability.

When we multiply $C_0$ by $D_0 x_0$, we will simply sum together the non-zero entries of $D_0 x_0$ that line up with the columns in $C_0^e$. For any $a$ that does not correspond to an active input, using the fact that $x_0$ has $O(\log m')$ non-zero entries and a union bound, we see that

$$\Pr[D_0 x_0(a) > 0] \le O\left(\frac{\log^2 m'}{n}\right) = O\left(\frac{\log m'}{\sqrt{m'}}\right).$$

Thus, the expected number of non-zero terms in the sum $C_0 D_0 x_0(e)$ is $O(\log m')$. Furthermore, since the entries of $D_0$ are chosen independently of each other, we can use a Chernoff bound to show

that the the number of non-zero terms in the sum for $C_0 D_0 x_0(e)$ is $O(\log m')$ with high probability. Finally, we point out that the incorrect non-zeros in $D_0 x_0$ have size at most $c/\log m'$ for a constant $c$ with high probability which follows directly from the facts that each entry of $D_0 x_0$ is the sum of $\log m'$ pairwise products of two entries, divided by $\log m'$ and the probability of each of those products being a 1 is at most $\log m'/n$. Putting all of this together shows that for any $e$, $C_0 D_0 x_0(e)$ is at most $O(1)$ with high probability. This can be made smaller than any $\epsilon$ by increasing $n$ by a constant factor dependent on $\epsilon$. $\square$

$\square$

## F.2 ALGORITHM FOR DOUBLE HEAVY OUTPUTS

We here provide the algorithm called **High-Influence-AND**, which is used by our high level algorithm for outputs that have two heavy inputs. Let $\bar{t}$ be the average influence of the feature circuit. The algorithm **High-Influence-AND** requires only $n = O(\sqrt{m'} \log m')$, provided that $\bar{t} > m'^{1/4}$. Note that the high level algorithm uses **High-Influence-AND** on a subproblem that has a *minimum* feature influence of $m^{1/4}$. This implies that $\bar{t} > m'^{1/4}$. However, **High-Influence-AND** applies more broadly than just when the minimum feature influence is high - it is sufficient for the *average* feature influence to be high. We here describe the algorithm in terms of the more general condition to point out that if the overall input to the problem meets the average condition, we can just use **High-Influence-AND** for the entire problem, instead of breaking it down into various subproblems.

This algorithm uses input channels, in the sense that the column specifications do not depend on which outputs the inputs appear in. We can do so here for all inputs, because the number of inputs $m$ is significantly smaller than $m'$, and we define $n$ relative to $m'$, not $m$. Specifically, if $\bar{t} > m'^{1/4}$, then $m' > m \cdot m'^{1/4}/2$, which implies that $m < 2m'^{3/4}$. This algorithm follows the same common structure as above, with the following modifications to $C_0$ and $D_0$:

---

**Algorithm 2 High-Influence-AND** (changes to $C_0$ and $D_1$)

1: **Construct $C_0 \in \{0,1\}^{n \times m}$ (one column per input):** For each input $j \in [m]$, sample the column $C_0(:,j)$ with i.i.d. Bernoulli($q$) entries where $q = m'^{-1/4}$. The $j$th column aligns with the $j$th coordinate of $D_0 x_0$. No additional columns are added to $C_0$.

2: **Define $D_1 \in \mathbb{R}^{m' \times n}$ by overlap of input codes:** For each output $i$ with input pair $(j_i, k_i)$, let

$$S_i := \{ t \in [n] : C_0(t, j_i) = 1 \wedge C_0(t, k_i) = 1 \}.$$

Set $D_1(i,t) = \frac{1}{|S_i|}$ for $t \in S_i$ and $D_1(i,t) = 0$ otherwise (each row averages over its overlap positions).

---

**Theorem F.4.** *With high probability Algorithm* **High-Influence-AND** *correctly computes $x_1$ from $x_0$, provided that at most 2 inputs are active, $\bar{t} > m'^{1/4}$, and $n = O(\sqrt{m'} \log m')$.*

*Proof.* We first point out that for any output that should be active as a result of the AND, the entries of $x_1$ that should be 1 for that output, will in fact be a 1. This follows from the fact that for any pair of inputs that appear in an output, the expected number of entries of overlap in their respective columns of $C_0$ is $\Theta(\log m')$, and thus we can use a Chernoff bound to show that it will be within a constant factor of that value. From there, we see that the correct value of $D_1 \text{ReLU}(C_0 D_0 x_0 + b)$ will be a 1. Thus, we only need to demonstrate that there is not too much noise of either type (a) or type (b) (as defined in Section F.1). We demonstrate this with the following two claims:

**Claim F.5.** *Any entry of $C'_1 y_1$ that does not correspond to a correct 1 of the 2-AND problem has value at most $\epsilon$ due to noise of type (b) with high probability.*

*Proof.* For any column of $C_0$, the expected number of 1 entries is $O(n/m'^{1/4}) = O(m'^{1/4} \log m')$, and will be no larger with high probability. With this in hand, we can use an argument analogous to that in the proof of Claim F.2. Specifically, for any entry $e$, Equation 1 still represents $E[C'_1 y_1(e)]$, and so it follows that $E[C'_1 y_1(e)] = \left( \frac{m'^{3/2} \log^3 m'}{n^3} \right)$. A similar Chernoff bound as in Claim F.2

shows that $C'_1 y_1(e)$ will be within a constant of its expectation with high probability. Thus, $n = O(\sqrt{m'} \log m')$ is sufficient to make $C'_1 y_1(e) \leq \epsilon$ with high probability. $\qquad\square$

**Claim F.6.** *The amount of type (a) noise introduced to any entry of $C_0 D_0 x_0$ is at most $\epsilon$ with high probability.*

*Proof.* Let $N(e)$ be the contribution to entry $e$ in $C_0 D_0 x_0$ due to this kind of noise. We first provide an expression for $E[N(e)]$. Let $H(C_0)$ be the columns of $C_0$, except those that correspond to the active inputs. In this expression, we let $a$ range over the columns of $H(C_0)$ and $b$ range over all the columns of $D_0$. We see that

$$E[N(e)] \;=\; \frac{1}{\log m'} \sum_{a \in H(C_0)} \sum_{b \in 1 \ldots n} \Pr[x_0(b) \;=\; 1] \Pr[D_0(a,b) \;>\; 0] \Pr[C_0(e,a) \;=\; 1], \quad (2)$$

where the active input not being in $H(C_0)$ implies that the three probabilities listed are independent. Since $|H(C_0)| \leq m$, there are at most $nm$ terms in this sum, and the first two probabilities are $\frac{\log m'}{n}$, and the third is $\frac{1}{m'^{1/4}}$. This gives us that

$$E[N(e)] = O\left( \frac{1}{\log m'} nm \left( \frac{\log m'}{n} \right)^2 \frac{1}{m'^{1/4}} \right) = O\left( \frac{m \log m'}{n m'^{1/4}} \right) = O(1),$$

where the last equality uses the fact that $m \leq 2m'^{3/4}$, which follows from the fact that $\bar{t} \geq m'^{1/4}$. This gap between $m$ and $m'$ is why we are able to use this algorithm in the case of high average feature influence, but not when that average is smaller. Since the summation of probabilities is divided by a $\log m'$ factor, a fairly straightforward Chernoff bound over the choices of $C_0(e,a)$, for $a \in H(C_0)$, shows that this is no higher than its expectation by a constant factor with high probability. The resulting constant can be made smaller than any $\epsilon$ by increasing $n$ by a constant factor dependent only on $\epsilon$. $\qquad\square$

This concludes the proof of Theorem F.4. $\qquad\square$

### F.3 Algorithm for mixed outputs

We now turn to the most challenging of our three subproblems, the case where the outputs are mixed: one heavy and one light input. As stated above, **High-Influence-AND** from Section F.2 is actually effective when some outputs are mixed, provided that the average feature influence of the feature circuit is sufficiently high. However, what **High-Influence-AND** is not able to handle (with $n = O(\sqrt{m'} \log m')$), is the case where the feature circuit has low average influence, but high maximum influence. Our algorithm here is used by the high level algorithm for all the mixed outputs, but most importantly it addresses that case of feature circuits with low average influence and high maximum influence. This involves a combination of input channels for heavy inputs and output channels for light inputs. Furthermore, we see below that just how high the feature influence of a heavy input is impacts how the problem is divided into input and output channels. As a result, we will further partition the outputs into two subcases based on a further refinement of the heavy features. Since we overall performed four partitions, this does not affect the overall asymptotic complexity of the solution.

In the case of regular heavy inputs, we can view the heavy inputs as using input channels (since they are not dependent on how those inputs are used), and the light inputs as using output channels (since they are routed to the channel of the input they share an output with). We see below that this is effective for regular heavy inputs. However, for super heavy inputs, this would not work: a super heavy input would have too many light inputs routed to it. If we do not increase the size of the input channel for the super heavy input, there will be too many light inputs routed to too little space, and as a result, those light inputs would create too much type (a) noise. And if we do increase the size of the input channel for super heavy inputs, then the super heavy inputs will create too much type (b) noise with each other. Thus, we need to deal with the super heavy inputs separately, as we did above. Key to this is **detect-two-active-heavies** which is a way of shutting down this entire portion of the algorithm when two super heavy inputs are active. This allows us to remove what would otherwise

---

**Algorithm 3 Mixed-Influence-AND** (mixed outputs; *changes only*, otherwise use common structure)

---

1: **Identify super-heavy inputs:** Label any input that appears in more than $m'^{1/2}$ outputs as *super heavy*. Partition the mixed-output handling into two cases: (i) regular heavy mixed outputs and (ii) super-heavy mixed outputs.

2:

3: *For regular heavy mixed outputs:*

4: **Disjoint encoding for light vs. super heavy:** In the encoding of $x_0$ and the matrix $D_0$, allocate disjoint rows/columns to light inputs and to super-heavy inputs so they do not share any rows or columns.

5: **Columns of $C_0$ for heavy inputs:** For each heavy input, add one column to $C_0$ whose entries are i.i.d. Bernoulli$(m'^{-1/4})$.

6: **Columns of $C_0$ for light inputs conditioned on co-appearance:** For each light input $j$, add one column to $C_0$. For entry $k$ in this column: if there exists a heavy input $i$ that co-appears with $j$ in some output and the heavy column $i$ has a 1 at entry $k$, then set entry $k$ for $j$ by an independent Bernoulli$(m'^{-1/4})$ draw; otherwise set it to 0.

7: **No other $C_0$ columns:** Do not allocate additional columns in $C_0$.

8: **Nonzeros of $D_1$ (gating rule):** For an output, the nonzero entries of its row in $D_1$ are exactly the positions where *both* of its two input columns in $C_0$ have a 1 (as in **Low-Influence-AND/High-Influence-AND**).

9:

10: *For super-heavy mixed outputs:*

11: **Stronger disjointness, including among super-heavies:** In $x_0$'s encoding and $D_0$, allocate disjoint rows/columns to light inputs and to super-heavy inputs, and additionally ensure distinct super-heavy inputs do not share rows or columns with each other.

12: **Columns of $C_0$ for heavy inputs:** For each heavy input, add one column to $C_0$ with i.i.d. Bernoulli$(\gamma^{-1})$ entries, for a constant $\gamma$ (chosen below).

13: **Columns of $C_0$ for light inputs:** For each light input, add one column to $C_0$ with i.i.d. Bernoulli$(2\gamma/\sqrt{m'})$ entries.

14: **Auxiliary mechanism:** Include a subroutine **detect-two-active-heavies** (described separately) to handle the super-heavy regime.

15: **Nonzeros of $D_1$ (gating rule):** As above, for each output, keep exactly those positions where both participating input columns in $C_0$ have 1s.

---

be too much noise in the system. The output for that pair of inputs will instead be produced by Algorithm **High-Influence-AND**.

**Theorem F.7.** *With high probability, Algorithm **Mixed-Influence-AND** correctly computes $x_1$ from $x_0$, provided that at most 2 inputs are active, each output contains both a heavy and a light input, and $n = O(\sqrt{m'}\log m')$.*

*Proof.* This follows from the two lemmas below.

**Lemma F.8.** *The subproblem of Algorithm **Mixed-Influence-AND** on the regular heavy mixed outputs produces the correct result provided that at most 2 inputs are active and $n = O(\sqrt{m'}\log m')$.*

*Proof.* We first point out that for any output that should be active as a result of the AND, the entries of $x_1$ that should be 1 for that output, will in fact be a 1. This follows from the fact that for any pair of inputs that appear in a regular heavy mixed output, the expected number of rows of overlap in their respective columns of $C_0$ is $\Theta(\log m')$, and thus we can use a Chernoff bound to show that it will be within a constant factor of that value. The Lemma now follows from the following two claims:

**Claim F.9.** *Any entry of $C_1'y_1$ that does not correspond to a correct 1 of the 2-AND problem has value at most $\epsilon$ due to noise of type (b) with high probability.*

*Proof.* For any column of $C_0$ (corresponding to either a light or a heavy input), the expected number of 1 entries is $O(n/m'^{1/4}) = O(m'^{1/4} \log m')$, and will be no larger with high probability. With this in hand, we can use an argument analogous to that in the proof of Claim F.2. $\square$

**Claim F.10.** *The amount of type (a) noise introduced to any entry of $C_0 D_0 x_0$ is at most $\epsilon$ with high probability.*

*Proof.* We need to argue this for both the light inputs and the heavy inputs. However, since we partitioned those inputs in $D_0$, they will not interfere with each other, and we can handle each of those separately. We first examine the heavy inputs, and note that there can be at most $m'^{3/4}$ of them, since each will contribute at least $m'^{1/4}$ distinct outputs. Let $N_h(e)$ be the contribution to $C_0 D_0 x_0(e)$ of this kind of noise from heavy inputs. We first provide an expression for $E[N_h(e)]$. Let $H(C_0)$ be the columns of $C_0$ in row $e$ that correspond to heavy inputs, not counting the active input. Let $H(D_0)$ be the columns of $D_0$ that are used by the heavy inputs. In this expression, we let $a$ range over the columns in $H(C_0)$ and $b$ range over the columns of $H(D_0)$. We see that

$$E[N_h(e)] = \frac{1}{\log m'} \sum_{a \in H(C_0)} \sum_{b \in H(D_0)} \Pr[x_0(b) = 1] \Pr[D_0(a,b) > 0] \Pr[C_0(e,a) = 1], \quad (3)$$

where the active input not being in $H(C_0)$ implies that the three probabilities listed are independent. Since $|H(C_0)| \leq m'^{3/4}$ and $|H(D_0)| \leq n$, there are at most $nm'^{3/4}$ terms in this sum, and the first two probabilities are $\frac{\log m'}{n}$, and the third is $\frac{1}{m'^{1/4}}$. This gives us that

$$E[N_h(e)] = O\left(\frac{1}{\log m'} nm'^{3/4} \left(\frac{\log m'}{n}\right)^2 \frac{1}{m'^{1/4}}\right) = O(1).$$

Since the summation of probabilities is divided by a $\log m'$ factor, a fairly straightforward Chernoff bound over the choices of $C_0(e,a)$, for $a \in H(C_0)$, shows that this is no higher than its expectation by a constant factor with high probability. The resulting constant can be made smaller than any $\epsilon$ by increasing $n$ by a constant factor dependent only on $\epsilon$.

We next turn to light inputs. This is more challenging than the heavy inputs for two reasons. First, if we define $L(C_0)$ analogously to $H(C_0)$, then $|L(C_0)|$ can be larger than $m'^{3/4}$ because each light input appears in at most $m'^{1/4}$ outputs. It can be $\Theta(m)$, which means we would need to evaluate the sum in the expectation a different way. Second, the choices of $C_0(e,a)$, for $a \in L(C_0)$, are no longer independent, since those choices for two light inputs that share the same heavy input will both be influenced by the choice in row $e$ for that heavy input (see Figure 5). Thus the Chernoff bound to demonstrate the high probability result needs to be done differently. In fact, this lack of independence is why we need to handle the super heavy inputs differently in the algorithm. If, for example, there were a single heavy input $h$ that appeared in the same output as all of the light inputs, consider any row $e_h$ such that $C_0(e_h, h) = 1$. The expectation of $C_0 D_0 x_0(e_h)$ is $\Theta(m^{3/4}/n)$, which is too large. In other words, with such a super heavy input, even though the expectation $E[N_l(e)] = O(1)$ for every $e$, the distribution is such that with high probability there will be some $e_h$ such that $N_l(e_h) = \Theta(m^{3/4}/n)$.

Instead, we take a different approach here. For any row $e$ of $C_0$, and any set of columns $S$, let $C_0^S(e)$ be the set of entries $C_0(e,a)$ that are 1 for $a \in S$. We first show that with high probability, $|C_0^{L(C_0)}(e)| = O(\sqrt{m'})$. To do so, we demonstrate that the entries in $C_0$ in row $e$ for the heavy inputs leave at most $O(m'^{3/4})$ light inputs that make a choice for their entry in row $e$; the remainder are only in rows where all heavy inputs that appear with them have a 0 in row $e$, and thus they are set to 0 without making a choice. More precisely, for any heavy input $a$, let $\delta(a)$ be the set of light inputs that appear in an output with $a$. We wish to show that

$$\left| \bigcup_{a \in C_0^{H(C_0)}(e)} \delta(a) \right| = O(m'^{3/4}).$$

To do so, first note that $\sum_{a \in H(C_0)} \delta(a) \leq m'$, since there are at most $m'$ outputs, and each output has at most one light entry. We can now define random variables $z_a$ for each $a \in H(C_0)$, where

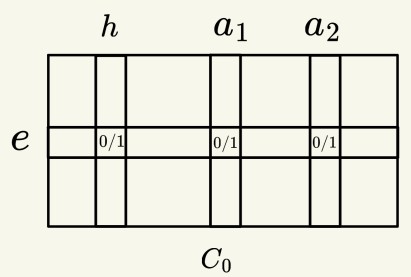

Figure 5: The dependence in light inputs between the different choices for $C_0(e, a)$, when $a \in L(C_0)$. Here $a_1$ and $a_2$ are light inputs, and so $a_1, a_2 \in L(C_0)$, and $h$ is a heavy input such that both $h \wedge a_1$ and $h \wedge a_2$ are computed. If we ignore the impact of other heavy inputs then if $C_0(e, h) = 0$, then both $C_0(e, a_1) = 0$ and $C_0(e, a_2) = 0$. Thus, $\Pr[C_0(e, a_2) = 1 | C_0(e, a_1) = 1] \gg \Pr[C_0(e, a_2) = 1 | C_0(e, a_1) = 0]$, and so $C_0(e, a_2)$ and $C_0(e, a_1)$ are not independent.

$z_a = 0$ when $a \notin C_0^{H(C_0)}(e)$, and $z_a = \delta(a)/\sqrt{m'}$ when $a \in C_0^{H(C_0)}(e)$, which happens with probability $\frac{1}{m'^{1/4}}$. Since there are no super heavy inputs in $H(C_0)$, $\forall a \in H(C_0), |\delta(a)| \leq \sqrt{m'}$, and so $0 \leq z_a \leq 1$. Also, the $z_a$ are mutually independent. Thus, $E\left[\sum_{a \in H(C_0)} z_a\right] \leq m'^{1/4}$, and a standard Chernoff bound shows that $\sum_{a \in H(C_0)} z_i = O(m'^{1/4})$ with high probability. From this it follows that

$$\left| \bigcup_{a \in C_0^{H(C_0)}(e)} \delta(a) \right| \leq \sum_{a \in C_0^{H(C_0)}(e)} |\delta(a)| = \sqrt{m'} \sum_{a \in H(C_0)} z_a = O(m'^{3/4}),$$

with high probability. Given this, at most $O(m'^{3/4})$ light inputs make a choice for their entry in row $e$, and each of those is a 1 independently with probability $\frac{1}{m'^{1/4}}$. A standard Chernoff bound now shows that $|C_0^{L(C_0)}(e)| = O(\sqrt{m'})$ with high probability.

To finish the proof of this claim, let $N_l(e)$ and $L(D_0)$ be defined analogously to $N_h(e)$ and $H(D_0)$ respectively, for light inputs. We see that

$$E[N_l(e)] = \frac{1}{\log m'} \sum_{a \in C_0^{L(C_0)}(e)} \sum_{b \in L(D_0)} \Pr[x_0(b) = 1] \Pr[D_0(a, b) > 0] = O\left( \frac{n\sqrt{m'}}{\log m'} \left( \frac{\log m'}{n} \right)^2 \right) = O(1).$$

We can now use a Chernoff bound over the choices of the relevant entries in $D_0$ to show that $N_l(e) = O(1)$ with high probability as well. The resulting constant can be made smaller than any $\epsilon$ by increasing $n$ by a constant factor dependent only on $\epsilon$. $\qquad\square$

This concludes the proof of Lemma F.8. $\qquad\square$

**Lemma F.11.** *The subproblem of Algorithm* **Mixed-Influence-AND** *on the super heavy mixed outputs produces the correct result provided that at most 2 inputs are active and* $n = O(\sqrt{m'}\log m')$.

*Proof.* We first point out that for any output that should be active as a result of the AND, the entries of $x_1$ that should be 1 for that output, will in fact be a 1. This follows from the fact that for any pair of inputs that appear in a super heavy mixed output, the expected number of rows of overlap they share in $C_0$ is $\Theta(\log m')$, and thus we can use a Chernoff bound to show that it will be within a constant factor of that value. The Lemma now follows from the following two claims:

**Claim F.12.** *The amount of type (a) noise introduced to any entry of $C_0 D_0' y_1$ is at most $\epsilon$ with high probability.*

*Proof.* Since the super heavy inputs do not have any overlapping columns in $D_0$ with each other or with light inputs, none of the super heavy inputs will produce type (a) noise. Note that there can be at most $\sqrt{m'}$ super heavy inputs (or we would have more than $m'$ outputs), and so $n = O(\sqrt{m'} \log m')$ is sufficient space to provide a non-overlapping input encoding in $x_0$ for each super heavy input. (This is why we cannot treat regular heavy inputs the same as super heavy inputs - there might be too many of them.) Thus, we only need to concern ourselves with type (a) noise produced by pairs of inputs that are both light. Demonstrating that this noise is at most $\epsilon$ is analogous to the proof that there is not too much type (a) noise for heavy inputs in Claim F.10. In fact, the expected such noise is given by an expression almost identical to Equation 3. In evaluating that expression, we only need to change the number of choices of $H(C_0)$ from $m'^{3/4}$ to $m$, and the $\Pr[C_0(e, a) = 1]$ from $\frac{1}{m'^{1/4}}$ to $\frac{2\gamma}{m'^{1/2}}$. The facts that the expectation of this noise is $O(1)$, that it is not much higher with high probability, and that it can be made smaller than $\epsilon$ by increasing $n$ by a constant all follow the same way as in the proof of Claim F.10. $\square$

**Claim F.13.** *When the two active inputs to the 2-AND problem consist of at most one super heavy input, then any entry of $C_1'y_1$ that does not correspond to a correct 1 of the 2-AND problem has value at most $\epsilon$ due to noise of type (b) with high probability.*

*Proof.* Type (b) noise occurs when the 1s that appear in $ReLU(C_0 D_0 x_0)$ are picked up by non-zero entries in unintended rows during the multiplication by $D_1$, and then remain after being subsequently multiplied by $C_1'$. Since we assume there is at most 1 super heavy input, we only need to handle two cases: one active light input and one active super heavy input, as well as two active light inputs. For two light inputs, the number of 1s in $ReLU(C_0 D_0 x_0)$ is $O(1)$ with high probability. For the mixed case, the number of 1s in $ReLU(C_0 D_0 x_0)$ is $O(\log m')$ with high probability, and thus is more challenging, and in fact the two active light inputs case can be handled similarly, so we here only present the argument for the mixed case.

Let $s$ be the active super heavy input, and let $l$ be the active light input. The 1s in $ReLU(C_0 D_0 x_0)$ from those two active inputs can be picked up by an unintended row of $D_1$ that corresponds to an incorrect output that combines a light input $l'$ and a heavy input $s'$, where either $s' \neq s$ or $l' \neq l$, or both. We will combine these three possibilities into two cases: in the first, $s' \neq s$, but $l' = l$, and in the second, $l' \neq l$, but $s'$ may or may not be the same as $s$.

In the first case, the number of entries of $D_1$ in the row for any given output that overlap with 1s, for each $s'$, can be at most $O(\log m')$, and since $l$ is light, there can be at most $m'^{1/4}$ such $s'$. Thus, this way only contributes at most $O(m'^{1/4} \log m')$ nonzero entries to $y_1 = D_1 ReLU(C_0 D_0 x_0)$. Furthermore, each of these entries has size at most $2/\gamma$ with high probability. The type (b) noise of any entry $e$ of $C_1'y_1$ will consist of the sum of each of those entries multiplied by either 0 or a 1, with the probability of a 1 being $\log m'/n$. Thus, from a union bound the probability that this sum is non-zero is at most $O(m'^{1/4} \log^2 m'/n) = O(\log m'/m'^{1/4})$. Furthermore, with high probability that sum will have at most $O(1)$ non-zero entries, and thus the type (b) noise when we hold $l$ fixed is at most $O(1)$, and that constant can be made smaller than any $\epsilon$ by increasing the constant $\gamma$.

We next turn to the second case: noise of type (b) that combines $s'$ with $l'$, where $l' \neq l$. In this case, the number of entries of $D_1$ in the row for any given output that overlap with 1s will be $O(1)$ with high probability, and thus any non-zero entry of $D_1$ has value $O(\frac{1}{\log m'})$ with high probability. There are at most $m'$ rows of $D_1$ that could have such overlap, and the probability of overlap for each of them is $O\left(n(\frac{1}{\sqrt{m'}})^2\right) = O\left(\frac{\log m'}{\sqrt{m'}}\right)$. Thus the resulting expected number of entries of $y_1$ that are non-zero is $O(\sqrt{m'} \log m')$. Using the fact that for any pair of rows of $D_1$ that involve two different light inputs, the entries in those rows will be independent, and the fact that every light input can appear in at most $m^{1/4}$ rows, we can use a Chernoff bound to show that it will not be higher by more than a constant factor.

Again, any entry $e$ of $C_1'y_1$ will consist of the sum of each of those entries multiplied by either 0 or a 1, with the probability of a 1 being $\log m'/n$. The expected number of non-zero terms in that sum will be $O\left(\frac{\log^2 m'}{n\sqrt{m'}}\right) = O(\log m')$, and can be shown with a Chernoff bound to be within a constant factor of its expectation with high probability. Finally, since each of these terms is $O(\frac{1}{\log m'})$ with

high probability, we see that this contribution to any entry of $C_1' y_1$ is at most $O(1)$. This can be made smaller than any $\epsilon$ by increasing $n$ by a constant factor. $\qquad \square$

Claim F.13 assumes that no two super heavy inputs are active. However, as described thus far, if two super heavy inputs were to be active, than a constant fraction of the entries in $ReLU(C_0 D_0 x_0)$ would be 1s, and this would wreak havoc with the entries in $C_1' y_1$. Fortunately, we do not need to handle the case of two active super heavy inputs here: if an output has two super heavy inputs, it will be handled by algorithm **High-Influence-AND**. However, we still have to ensure that when there are two active super heavy inputs, all of the mixed outputs return a 0. Specifically, when there are two active super heavy inputs, there is so much noise of type (b) that if we do not remove that noise, many mixed outputs would actually return a 1. The mechanism **detect-two-active-heavies** is how we remove that noise.

To construct this mechanism, we add a single row to the matrix $C_0$, called the cutoff row. Every column of $C_0$ corresponding to a super heavy input will have a 1 in the cutoff row, and all other columns will have a 0 there. There will be a corresponding bias of $-1$ that lines up with this entry, and so the cutoff row will propagate a value of 1 if there are two or more super heavy inputs and 0 otherwise. $D_1$ will have a cutoff column which lines up with the cutoff row in $C_0$, and that column will have a value of $-Z$ in every row, where $Z$ is large enough to guarantee that all entries of $y_1$ will be negative. Thus, all entries of $C_1' y_1$ will be non-positive, and will be set to 0 by the subsequent $ReLU$ operation. Note that since we are using non-overlapping entries of $x_0$ to represent the super heavy inputs, there will not be any noise added to the cutoff row, and so this mechanism will not be triggered even partially when less than two super heavy inputs are active.

We point out that this operation is very reliant on there being at most two active inputs, and so the algorithm as described thus far does not work if three or more inputs are active (for example, two super heavy inputs and one light input would only return zeros for the mixed outputs). However, we describe below how to convert any algorithm for two active inputs into an algorithm that can handle more than two active inputs. $\qquad \square$

This concludes the proof of Theorem F.7. $\qquad \square$

## G  GENERALIZING THE CONSTRUCTIONS

We here demonstrate how the above algorithm can be made efficient in terms of the number of bits required to represent the parameters, and also how it can be extended to more general settings, adding the ability to structurally handle more than two active inputs, handle multiple layers, and handle the $k$-AND function.

### G.1  BIT COMPLEXITY OF PARAMETERS

We describe how to ensure the algorithm can be constructed using an average of $O(1)$ bits per parameter while maintaining computational correctness. First, observe that the compression matrices ($C$, $C_0$, and $C_1'$ across all of the cases we consider) contain only binary entries, which allows them to be represented efficiently. The decompression matrices ($D$, $D_0$, and $D_1$), however, contain values between 0 and 1 determined by a normalizing term from the corresponding columns of the $C$ matrices. But, since the algorithm thresholds its final results, exact normalization values are unnecessary. In all cases, the required normalization term is $\Theta(1/\log m')$ w.h.p., and can be approximated with a single representative value $\nu = c/\log m'$ for a suitably chosen constant $c$.

This approximation ensures that the representation of $C$ and $D$ only requires us to use an encoding of three different values for each entry: 0, 1, and $\nu$. However, the final protocol is obtained by multiplying these matrices together to obtain the $n \times n$ matrices of the form $CD$, and so we need to understand the entries in these product matrices. An analysis (not included here) of all of the different matrices utilized shows that each entry in any $CD$ can be modeled as a random variable $\nu\rho$, where $\rho$ is the sum of $r$ independent Bernoulli$(1/r)$ random variables, where $r$ ranges in the different protocols between $\sqrt{m}$ and $m'$ (with a minor modification for the algorithm **Mixed-Influence-AND**, which involves two such sums).

To encode these entries efficiently, we only need to encode the value of $\rho$, and we do so with essentially a unary code: $\rho$ is represented by a string of $\rho - 1$ "1" symbols followed by a "0" symbol. The probability distribution of the value of $\rho$ is such that the expected number of bits this requires is $O(1)$: we are effectively using a Huffman code on a set of events $\rho_1, \ldots, \rho_r$, where $\Pr[\rho_{i+1}] \leq \Pr[\rho_i]/2$. Thus, each entry of any $CD$ matrix can be represented using an expected $O(1)$ bits, confirming the desired average $O(1)$-bit complexity per parameter.

We point out that this does require the model to "unpack" these representations at inference time. If we require the parameters to be standard real number representations, then we instead use the fact that for all matrix entries, $\rho = O(\log m')$ w.h.p. (and if we are unlucky with our random choices and it is larger, we can recreate the algorithm from scratch until we achieve this result). As a result, all values in all matrices will be $O(1)$, and it is sufficient for us to represent them with a precision of $O(1/\log m')$. Thus, for this more stringent requirement on representation, $O(\log \log m')$ bits per parameter is sufficient.

## G.2 More than two active inputs

We have assumed throughout that at most 2 inputs are active at any time. It turns out that most of the pieces of our main algorithm work for any constant number of inputs being active, but one significant exception to that is the **detect-two-active-heavies** mechanism of Algorithm **Mixed-Influence-AND**, which requires at most 2 active inputs in order to work. Thus, we here describe a way to handle any number of active inputs, albeit at the cost of an increase in $n$. Let $v$ be an upper bound on the number of active inputs.

We start with the case where $v = 3$, where there are three possible pairs of active inputs to a 2-AND. The idea will be to create enough copies of the problem so that for each of the three possible pairs of active inputs, there is a copy in which the pair appears without the third input active. To handle that, we make $O(\log m)$ copies of the problem (and thus increase $n$ by that factor). These copies are partitioned into pairs, and each input goes into exactly one of the copies in each pairing. The choice of copy for each input is i.i.d. with probability 1/2. Each of the copies are now computed, using our main algorithm, except that only the outputs that have both of their inputs in a copy are computed, and we have an additional mechanism, similar to **detect-two-active-heavies**, that detects if a copy has 3 active inputs, and if so, it zeroes out all active outputs in that copy. Finally, we combine all the copies of each output that are computed, summing them up and then cutting off the result at 1. The number of copies is chosen so that for every set of three inputs, with high probability there will be a copy where each of the three possible pairs of inputs in that set of three inputs appears without the third input. Thus, for any set of three active inputs, each pair will be computed correctly in some copy, and so with high probability this provides us with the correct answer.

We can extend this to any bound $v$ on the number of active inputs. We still partition the copies into pairs, and we need any set of $v$ inputs to have one copy where each set of two inputs appears separately from the other $v - 2$ inputs. The probability for this to happen for a given set of $v$ inputs and a pair within that set is $1/2^{v-1}$. The number of choices of such sets is $\binom{m}{v}\binom{v}{2}$. Thus, to get all of the pairings we need to occur, the number of copies we need to make is $O\left(2^{v-1} \log\left[\binom{m}{v}\binom{v}{2}\right]\right) \leq O(v2^v \log m)$. As a result, we can still compute in superposition up to when $v = O(\log m')$. We note that some care needs to be taken with the initial distribution of copies of each input to ensure that process does not create too much (type (a)) noise, but given how quickly $n$ grows with $v$ due to the number of copies required, this is not difficult.

## G.3 Multiple layers

These algorithms can be used to compute an unlimited number of layers because, as discussed above, as long as the output of a layer is close to the actual result (intended 1s are at least 3/4 and intended 0s are at most 1/4), we can use thresholding to make them exact Boolean outputs. Therefore, the noise introduced during the processing of a layer is removed between layers, and so does not add up to become a constraint on depth. Also, as discussed, the high probability results all are with respect to whether or not the algorithm for a given layer works correctly. Therefore, each layer can be checked for correctness, and redone if there is an error, which ensures that all outputs are computed correctly for every layer. Thus, there is no error that builds up from layer to layer.

### G.4 Computing $k$-AND

We next turn our attention to the question of $k$-AND. To do so, we simply utilize our ability to handle multiple layers of computation to convert a $k$-way AND function to a series of pairwise AND functions. Specifically, to compute each individual $k$-AND, we build a binary tree with $k$ leaves where each node of the tree is a pairwise AND of two variables. These then get mapped to a binary tree of vector 2-AND functions where each individual pairwise AND is computed in exactly one vector 2-AND function, to compute the entire vector $k$-AND function. We thus end up with $2k$ 2-AND functions to compute, which we do with an additional factor of $\log k$ in the depth of the network.

As described so far, this will increase the number of neurons required by the network by a factor of $O(k)$. However, for $k$-AND to be interesting, there would need to be the possibility of at least $k$ inputs being active ($v \geq k$), and so for any interesting case of $k$-AND, we would be using our technique for more than 2 active inputs described above, and so if we want to compute in superposition, we have the limitation that $k = O(\log m')$. However, since that technique already ensures that every pair of the $k$-AND appears by itself in one of the copies of the network, we can embed the leaves of our binary tree into those copies. We would then do the same thing with the next level of the tree, and so on until we get to the root of the tree. Since each level of the tree has half the number of outputs as the previous level, we get a telescoping sum, and thus $k$-AND can be added to our implementation of at least $k$ active inputs without changing the asymptotics of $n$. It does however add an additional factor of $\log k$ to the depth of the network.

