# OpenReview forum: "On the Complexity of Neural Computation in Superposition"
_ICLR.cc/2026/Workshop/Sci4DL — Sci4DL 2026_

### Official Review · Reviewer_xmDU · 2026-02-24

**Fit:** 1
**Significance:** 1
**Confidence:** 2

**Summary:**

This paper establishes lower and upper complexities on the number of neurons and parameters required to compress information with superposition in a general neural class of functions.

**Strengths:**

The paper is very clear in the stated problem setting in the main body. The differentiation from existing literature is stated clearly, and the conclusions are quite profound; the generality of the models in which it holds is also a nice result.

**Suggestions:**

Although a strong theoretical paper, I do not believe this paper to be relevant to the workshop. The workshop's stated purpose is to "promote a complementary approach" to proving "rigorous theorems about optimization or generalization errors of standard algorithms...," but the paper takes this approach.  It does not empirically study any design choices, in line with "empirical analyses of deep networks that can validate or falsify existing theories and assumptions, or answer questions about the success or failure of these models" called by the workshop.

 The organization is not formatted into reader-friendly sections.

---

### Official Review · Reviewer_bfPu · 2026-02-26

**Fit:** 3
**Significance:** 2
**Confidence:** 2

**Summary:**

This paper looks at the ability of feedforward ANNs to compute in "superposition", where the number of features that a network can compute exceeds the number of neurons. The work includes technical proofs in an appendix and shows that the number of features can be at most nearly quadratic in the number of neurons. Some specific results are given for representing pairwise-AND functions

**Strengths:**

The paper is readable and the parts of the appendix I read seemed correct, although the appendix was much too long to check all of the proofs.

I think that people will be interested in these results.

**Suggestions:**

The work is well-suited for this workshop, but overall should be published in a journal or full-form paper where all the results can be checked.

Some experiments to support the theory would be helpful, especially to show that the explicit construction used for the upper bounds (App C) succeeds.

A formal definition of what is a "feature" should be given at some point. I'm not sure what you exactly mean.

* Including section C.1 in the main text might make more clear how the construction works
* line 120: calling it "v-sparse" might be more readable than "feature sparse v"
* Theorem E.1: the proof talks about "many functions" from $\mathcal{F}$, but this is too vague for a proof. What do you mean when you say many functions? I think you mean an average.

---

### Meta-Review · Area_Chair_Bsqu · 2026-02-28

**Recommendation:** Accept

**Metareview:**

Reviewers found the paper to be of good quality. One doubted its relevance to this workshop. I share these doubts. Nonetheless, science is a multi-stage process, and it's conceivable that these results will play nicely with simple experiments. (Ideally, such expts should've been included for this workshop.) I'll thus make an exception.

---

### Decision · Program_Chairs · 2026-03-02

Accept